# Artificial neural network in the discrimination of lung cancer based on infrared spectroscopy

Eiron John Lugtu[1]*, Denise Bernadette Ramos[1], Alliah Jen Agpalza[1], Erika Antoinette Cabral[1], Rian Paolo Carandang[1], Jennica Elia Dee[1], Angelica Martinez[1], Julius Eleazar Jose[1], Abegail Santillan[2,3], Ruth Bangaoil[2,3,4], Pia Marie Albano[2,3,5], Rock Christian Tomas[6]

1 Department of Medical Technology, Faculty of Pharmacy, University of Santo Tomas, Manila, Philippines, 2 Research Center for the Natural and Applied Sciences, University of Santo Tomas, Manila, Philippines, 3 The Graduate School, University of Santo Tomas, Manila, Philippines, 4 University of Santo Tomas Hospital, Manila, Philippines, 5 Department of Biological Sciences, College of Science, University of Santo Tomas, Manila, Philippines, 6 Department of Electrical Engineering, University of the Philippines Los Baños, Laguna, Philippines

* ejllugtu@gmail.com

**Data Availability Statement:** All relevant data are within the manuscript.

**Funding:** The author(s) received no specific funding for this work.

## Abstract

Given the increasing prevalence of lung cancer worldwide, an auxiliary diagnostic method is needed alongside the microscopic examination of biopsy samples, which is dependent on the skills and experience of pathologists. Thus, this study aimed to advance lung cancer diagnosis by developing five (5) artificial neural network (NN) models that can discriminate malignant from benign samples based on infrared spectral data of lung tumors ($n = 122$; 56 malignant, 66 benign). NNs were benchmarked with classical machine learning (CML) models. Stratified 10-fold cross-validation was performed to evaluate the NN models, and the performance metrics—area under the curve (AUC), accuracy (ACC) positive predictive value (PPV), negative predictive value (NPV), specificity rate (SR), and recall rate (RR)—were averaged for comparison. All NNs were able to outperform the CML models, however, support vector machine is relatively comparable to NNs. Among the NNs, CNN performed best with an AUC of 92.28% ± 7.36%, ACC of 98.45% ± 1.72%, PPV of 96.62% ± 2.30%, NPV of 90.50% ± 11.92%, SR of 96.01% ± 3.09%, and RR of 89.21% ± 12.93%. In conclusion, NNs can be potentially used as a computational tool in lung cancer diagnosis based on infrared spectroscopy of lung tissues.

## Introduction

Lung cancer is considered the leading cause of death due to cancer worldwide [1] and has been the biggest cancer killer among men globally, and for women in countries such as North America, East Asia, Northern Europe, Australia, and New Zealand [2].

The initial evaluation of patients suspected of lung cancer starts with history taking and physical examination complemented with complete blood count and chest radiography [3, 4].

**Competing interests:** The authors have declared that no competing interests exist.

A negative result from the chest radiography, however, is not definitive as the location and size of the tumor, state of metastasis, and type of lung cancer must be checked as well [3, 5]. All patients who are eligible for treatment with the intention of curing the disease must be offered with computed tomography (CT) [4, 6], a positron emission tomography (PET) scan if necessary, and then a diagnostic evaluation shall be made [3]. CT scans must be done prior to any invasive procedures as it provides knowledge regarding anatomical changes which increase the diagnostic yield of investigation [4]. At present, Centers for Disease Control (CDC) only recommends low-dose computed tomography (LDCT) as the means of screening lung cancer [7]. However, this process is still known to provide false-positive results, leading to overdiagnosis [7], while also exposing the patients to low doses of radiation [7, 8].

Combinations of other testing are also employed. Magnetic resonance imaging (MRI), bronchoscopy, and histopathologic examinations are done as needed for diagnosis [9]. Although MRI is known to offer a non-invasive assessment without the radiation, it is still susceptible to cardiac and respiratory motion artifacts [9]. Meanwhile, bronchoscopy plays an important role in confirming diagnosis but it is dependent on tumor size and location [4]. The current gold standard, which is the microscopic examination of hematoxylin and eosin (H&E)-stained biopsies, would take about a week to complete [10]. Additionally, it is prone to interobserver variability, leading to diagnostic disagreement among pathologists which may affect the prognosis and future course of action [11–16]. In light of a disagreeing diagnosis, a genetic and/or molecular analysis may be requested for an even more definite diagnosis and treatment guidance [17].

Innovations to improve the diagnosis of lung cancer approach quickly. Attenuated total reflection Fourier transform infrared (ATR-FTIR) spectroscopy is one of the newer methods being ventured for a fast yet reliable diagnosis for cancer. It has been proven that ATR-FTIR can identify molecular fingerprints from different biofluids, thus making it a promising clinical diagnostic tool [18]. Consequently, it has been initially applied for the diagnosis of breast cancer and gynecological cancer, both of which have yielded high sensitivity and specificity values [19, 20]. Bangaoil *et al.* have also used it as an adjunct method in the assessment of H&E-stained biopsies of lung cancer, and it was concluded to be able to provide results that are at par with the gold standard [21].

Ergo, an axillary tool that has the combined characteristics of the aforementioned auspicious diagnostic measurement and the latest technological update may be utilized that would give birth to advancement in medical diagnostics. This would not only relieve the workload of medical professionals but could also provide a tool that is even more specific and sensitive with a shorter turn-around time.

Artificial intelligence (AI), more specifically deep learning (DL), has proven its ability over the past few years in several facets of everyday activities. These advancements of AI technology have also been integrated into several cancer studies, such as in thyroid cancer [22], ovarian cancer [23, 24], and breast cancer [25] to improve the accuracy and speed of diagnosis, ultimately delivering better healthcare services to patients. Its application in lung cancer diagnosis has also started. Most of which have been generally focused on the image analysis of either lymph nodes or pulmonary nodules [26–28]. One of the downsides, however, of image-based AI diagnostics is that it is heavily reliant on the abnormalities that are only visible on the scanned regions. This may cause a late diagnosis for the patient who might be already suffering from an advanced stage malignancy. Additionally, unstandardized protocols and procedures of different laboratories regarding dyes and other contrasting agents may become a hindrance in the training of the AI model.

This study then aims to combine the potential of both ATR-FTIR and AI in the diagnosis of lung cancer. FTIR-based AI gives an edge over image-based AI as it would only require

minimal sample amount and preparation [29] all while being able to capture small differences in detailed cell signatures [30], hence answering the limitations of image-based AI diagnostics. Additionally, the results generated by FTIR come at a smaller file size that is easier to process compared to images. Its training would then be faster, easier, and cost-efficient. The possibility of utilizing both FTIR spectroscopy and AI could secure an efficient diagnosis with high accuracy without compromising the patient's health by repeatedly exposing them to health hazards such as radiation.

Six (6) of the most widely used machine learning models were compared with five (5) designed NN models in terms of classifying spectral data of lung samples as malignant or benign. This method may serve as an efficient adjunct tool that could provide more insights for pathologists and medical practitioners upon diagnosis.

## Materials and methods

### Ethical clearance

The study of Bangaoil *et al*. was ethically cleared by the Institutional Review Boards (IRB) of both study sites, (1) University of Santo Tomas Hospital (USTH) in Manila, Philippines (Ref. No.: IRB-2017-09-191-IS) and (2) Mariano Marcos Memorial Hospital and Medical Center (MMMH-MC) in Ilocos Norte, Philippines (Ref. No.: MMMHMC-RERC-15-006) [21]. The current study was also ethically cleared by the Faculty of Pharmacy Ethics Committee (FOPREC) in Manila, Philippines (Reference No.: FOP-ERC-2021-01-014). The current study was limited to the use of the spectral data of the said specimen; thus, no written informed consent was applicable as well. No additional procedures were performed that may potentially cause a risk of harm to subjects. The current study was done in accordance with the fundamental principles of ethics and the Declaration of Helsinki.

### Study population and sample preparation

FTIR spectral data ($n$ = 122; 56 malignant, 66 benign) of FFPE lung biopsies from 112 adult patients seen at MMMH-MC and USTH from 2015 to 2017 comprised the dataset. No participants were recruited for this study as the spectral data were acquired from the previous study of Bangaoil *et al*. [21].

Sample preparation and pretreatment of specimens before ATR-FTIR analysis were also discussed in the previous study [21]. Three (3) adjacent sections were cut uniformly (5-μm thick) from the FFPE cell blocks using a microtome (Leica Biosystems, Germany) and then mounted on glass slides. The outer sections (2) were stained with H&E and distributed to two (2) external evaluators (pathologists) blinded of the original diagnosis for validation. The middle section was deparaffinized following standard protocols using xylol [31, 32], washed with water, and left to air dry overnight before spectral analysis [33]. The classification (i.e., whether spectral data was benign or malignant) was based on the microscopic examination of H&E-stained tissues from each study site.

Prior to spectral measurement, performance qualification (PQ) test protocols using the automated validation program of the OPUS 8.0 software were conducted. The deparaffinized tissue sections were placed and oriented directly in contact with the ATR diamond surface (2 mm x 2 mm). All 122 tissue sections were examined in the mid-IR region (4000 cm$^{-1}$ to 600 cm$^{-1}$) with an average of 48 scans added to obtain an adequate signal-to-noise ratio [34–36], yielding 122 spectral data with a resolution of 4 cm$^{-1}$. This is further supported by the software's validation program as "acceptable". Majority of the malignant samples were scanned entirely since the specimens were composed mostly of cancer cells. Otherwise, only the areas identified by the pathologists to contain cancer cells were scanned. For the benign tissue

sections, they were scanned randomly, covering 50% of the total area of the specimen. All specimens were scanned three (3) times for reproducibility. Spectral data in the fingerprint region (1800 cm$^{-1}$ to 850 cm$^{-1}$) were extracted and the median infrared spectra were calculated. The overall method implemented in the study is summarized in Fig 1, through a generalized process flowchart.

## Data measure/instrumentation

Bruker Alpha II Fourier Transform Infrared (FTIR) spectrometer (Bruker Optics, Germany) equipped with a platinum ATR single reflection monolithic diamond sampling module was used to acquire the infrared (IR) spectra. The fully automated validation program of OPUS 8.0 software (Bruker Optics, Germany) was used for the performance qualification (PQ) test. The same software was used for baseline correction of obtained IR spectra.

MATLAB R2020b (MathWorks, USA) was used to normalize all the spectral data, and to simulate the training, validation, and testing of all NN models and other machine learning models. All of the deep learning designs and classification models were coded from scratch to implement the design specifications of each model.

All models were primarily implemented on a personal computer workstation that has an AMD Ryzen 5 3600 6-core 12-thread processor, 3.60 GHz CPU, 16 GB RAM, 500 GB SSD hard drive, and an EVGA RTX 3070 XC3 ULTRA (California, United States).

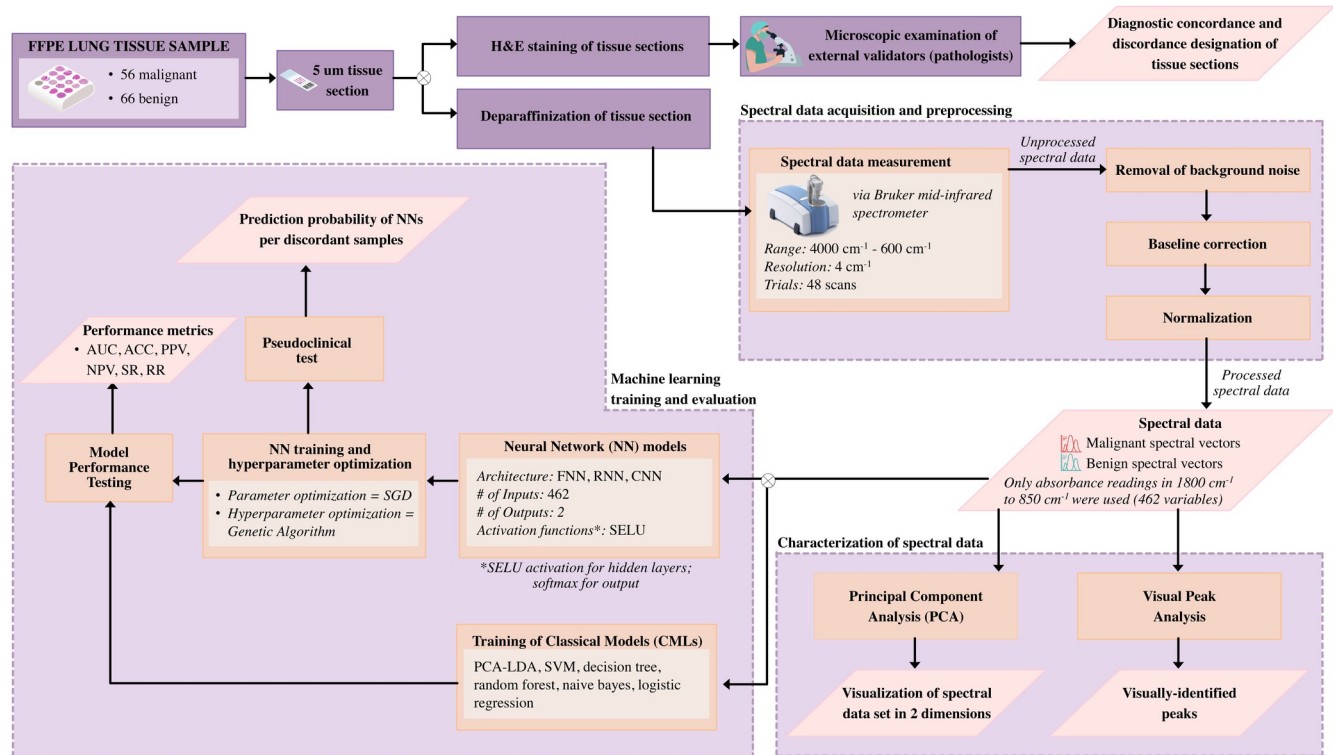

**Fig 1. Experimental design process flowchart.** The figure shows the experimental design of the study from acquisition of spectral data to machine learning training and evaluation.

## Pre-processing of spectral data

The spectral dataset $X_{SD}$ consisted of 122 elements which corresponded to 66 known benign and 56 known malignant samples. Each element $X_{SD}^{(i)} \in X_{SD}$ in the spectral data set consisted of 462 variables, which corresponded to the absorbance of a tissue sample for each wavenumber from 1800 cm$^{-1}$ to 850 cm$^{-1}$.

Normalization and baseline correction comprised the pre-processing of the spectral data set, $X_{SD}$. All spectral data were normalized using peak normalization before peak analysis and spectrum visualization. The normalization was performed to eliminate bias from y-value discrepancies among the IR samples due to environmental noise and instrument parameters. The peak normalization equation is as follows

$$X_{SD}^{(i)} = \frac{X_{SD}^{(i)} - \min\left(X_{SD}^{(i)}\right)}{\max\left(X_{SD}^{(i)}\right)} \ \forall \ sample \ i \tag{1}$$

where the $\min\left(X_{SD}^{(i)}\right)$ and the $\max\left(X_{SD}^{(i)}\right)$ terms refer to the minimum and the maximum absorbance of the $i^{th}$ spectral data, respectively. Note that the normalization was done per spectral data element/vector $X_{SD}^{(i)}$; hence the spectrum from other spectral vector does not influence the results of normalizing another spectral vector.

As regards the training process, the spectral data was normalized using z-score normalization since it is the recommended method of normalization for the scaled exponential linear unit (SELU)-based feed forward neural networks [37]. The normalization was done per spectral vector using the equation

$$X_{SD}^{(i)} = \frac{X_{SD}^{(i)} - mean\left(X_{SD}^{(i)}\right)}{std\left(X_{SD}^{(i)}\right)} \ \forall \ sample \ i \tag{2}$$

where the $mean\left(X_{SD}^{(i)}\right)$ and $std\left(X_{SD}^{(i)}\right)$ notations denote the mean and standard deviation of the elements of the vector $X_{SD}^{(i)}$. The implemented normalization scales the elements of $X_{SD}^{(i)}$ to have an overall mean of 0 and a standard deviation of 1. Note that similar to the peak normalization, the z-score normalization was performed per spectral vector $X_{SD}^{(i)}$.

Prior to the acquisition of spectral data, baseline correction and performance qualification tests had been conducted in the previous study [21]. The performance qualification tests included the signal-to-noise test, deviation from 100%-line test, interferogram peak test, and wavenumber accuracy test, all of which must be passed before ATR-FTIR analysis could be performed. $X_{SD}$ was processed using Opus 8.0 (Bruker Optics, Germany). Rubber-band baseline correction with 64 baseline points was performed to approximate a polynomial fit based on the minima of y-values in each spectral vector $X_{SD}^{(i)}$. The fitted polynomial was then deducted for all $X_{SD}^{(i)}$ to create the baseline-corrected spectrum [38–42]. Finally, the corrected spectrum was scaled within the fingerprint region, from 1800 cm$^{-1}$ to 850 cm$^{-1}$ [40, 43]. Other than baseline correction using the rubber band method, no further user intervention was done to assess the spectral data.

## Visual peak analysis

To characterize the data set, significant absorbance peaks in the fingerprint region were visually identified in $X_{SD}$. A test of normal distribution using the Shapiro-Wilk test was performed

for the identified peaks to decide whether a parametric or non-parametric test should be conducted. Since the spectral data set followed a non-normal distribution, they were subjected to the Mann-Whitney U test to further assess if the peaks of malignant samples were significantly different ($p$-value $< 0.05$) from that of benign samples. All the said statistical analyses were performed using MATLAB 2020b.

## Principal component analysis

To visualize the distribution of the spectral characteristic of the malignant and benign samples, $X_{PCA}$ was plotted in a principal component analysis (PCA) biplot using the two most dominant principal components, $F_1$ and $F_2$. The process of translating $X_{SD}$ to the reduced variable space, $X_{PCA}$ ($X_{SD} \rightarrow X_{PCA}$) is given by the equation

$$X_{PCA} = (X_{SD} - \bar{X}_{SD}) \times \left[ \vec{S}_{F1}^{\;T}, \vec{S}_{F2}^{\;T} \right] \qquad (3)$$

where $X_{PCA} \in \mathbb{R}^{N \times 2}$ is the reduced sample space, $\bar{X}_{SD}$ is the mean absorbance value of $\vec{x}_{SD}^{(i)} \in X_{SD} \forall\, i \leq N$, and $\vec{S}_{F1}$ and $\vec{S}_{F2}$ are the eigenvectors corresponding to the largest two eigenvalues of the covariance matrix $S_{SD} = (X_{SD} - \bar{X}_{SD})^{\;T} \times (X_{SD} - \bar{X}_{SD})$.

## Machine learning models

The most common machine learning models were utilized as classical benchmarks to compare the designed NN models in terms of diagnostic performance metrics. six (6) classification models were implemented in the study: linear discriminant analysis (LDA), support vector machine (SVM), logistic regression (LR), decision tree (DT), random forest (RF), and Naïve Bayes (NB).

Five (5) neural networks were designed and implemented, wherein three (3) were feed forward neural networks (FNN), one (1) single-layered recurrent neural network (RNN), and one (1) convolutional neural network (CNN). A Gaussian random initialization was performed to initialize the weights, while a zero-value initialization was done for the biases. All neural networks utilized scaled exponential linear unit (SELU) activation functions for all connections except for the output, which utilized a softmax function. Stochastic gradient descent was used to train and optimize the NN parameters where they were updated using adaptive gradient algorithm (AdaGrad). The cost function made use of the binary cross-entropy cost function. Genetic algorithm (GA) was used to optimize the NN hyperparameters.

**Linear discriminant analysis.** The LDA model was constructed following Fisher's Criterion [44], $J(w) = \frac{w^T S_B w}{w^T S_w w}$, where $S_B$ and $S_W$ denotes the between-class and the within-class covariance matrixes, derived from the training set $X_{TR} \in X_{SD}$, respectively; where $X_{TR}$ is a matrix of samples having 462 variables. Here, $w$ is the eigenvector with the highest eigenvalue derived from the solution of $\frac{dJ(w)}{dw}$ which is equal to $eig\left(S_C^{-1} S_B\right)$. The probability of malignancy, $p(X)$, for each sample $X^{(i)}$ was derived using the formula

$$p(X) = softmax([f_{norm}(s, \mu_B, \sigma),\, f_{norm}(s, \mu_M, \sigma)]) \mid s = X \times w \qquad (4)$$

where the function $f_{norm}$ denotes the value of a Gaussian probability density function at a point $s$, with means $\mu_B$ and $\mu_M$ which are the benign and the malignant mean projected values, respectively, from the training set, and $\sigma$ is the linear distance between $\mu_B$ and $\mu_M$. The process was repeated for 50 trials to ensure stability.

**Support vector machine.** The designed support vector machine (SVM) is a linear SVM. The SVM is designed using the elements of the training set $X_{TR} \in X_{SD}$, by considering an

unconstrained Lagrange optimization problem [45, 46]. To determine sub-optimal values for the parameters of the SVM, stochastic gradient descent (SGD) was used. A grid search was performed to select the best learning rate for the SGD; the validation set accuracy was considered as the optimization metric which determined the superiority of one model over the other. The output probability diagnosis of the model for benign and malignant cases was computed using Platt's method [47]. The process was repeated for 50 trials to ensure stability.

**Logistic regression.** The designed logistic regression (LR) model is a 462-input classifier with an output probability quantifying the likelihood of a sample to be malignant. In training the model, SGD was used over a training set, $X_{TR} \in X_{SD}$ using the binary cross-entropy function as the loss function. A grid search was performed, similar to that of the SVM, to optimize the SGD learning rate. The process was repeated for 50 trials to ensure stability.

**Decision tree and random forest.** The classification and regression trees (CART) algorithm were used to generate decision trees (DTs) of binary splits. The Gini's diversity index [48] was used to find the best input variable ($\omega = X^T \mid \omega \in \mathbb{R}^{462 \times N}$) for splitting the training set for each iteration of branching. Since the values of $\omega$ are continuous, the best value of separation was identified by considering the variable $\omega_j \in \omega$ having the least *Gini* metric [49]. The branching was recursively performed for each newly created node until the performance in the validation set accuracy decreased. The process was repeated for 50 trials to ensure stability.

The designed random forest (RF) model utilized the creation of trees following the previously discussed. The diagnosis of the RF was determined as the prevailing diagnosis made by its constituent bags of DTs. To determine the optimum number of trees $N_{RF}$ for the RF, a grid search from 3 to 100 trees was performed. The validation set accuracy was considered as the optimization criteria of the search. Each simulation was repeated over 50 times for each iteration to ensure stability. The average performance metric over the 50 trials served as the final performance metric of the RF with the number of trees set to $N_{RF}$.

**Naïve bayes.** The designed NB is a classifier of 2 classes. For each input variable $\omega_j \in \omega$, the best value of separating the two sub-classes was determined. The algorithm for finding the best value of separation is the same as that of the DT and RF designs where the Gini's index was used. The NB classifier outputs a malignant diagnosis when the probability obtained was more than 50%, otherwise the diagnosis is benign. In order to determine the optimal number of input for the classifier, the number of inputs was increased from 3 to 462, where the inputs of the least $GINI(j)_t$ value were considered first. The optimization was terminated at the iteration where the validation accuracy of the model started to decrease. Each iteration of was repeated for 50 trials, where the average validation accuracy from the 50 trials was the considered optimization metric criterion. The final NB constituted to the design with the highest average validation accuracy.

**Feed forward neural network design.** Three (3) FNN models were designed with varying layer sizes ($N = 2$, $N = 4$, $N = 8$). The number of neurons per layer was kept constant for every hidden layer. Each hidden layer $h_i$ made use of a 90% *SELU* dropout as recommended by SELU-based NNs [37]. Fig 2 shows the general architecture of the designed FNNs.

**Recurrent neural network design.** The designed RNN classifier made use of a recurrent neural network (RNN) base, coupled with a single-layered FNN at its output layer. The designed RNN architecture is illustrated in Fig 3, where an unrolled RNN is shown.

**Convolutional neural network design.** The designed convolutional neural network (CNN) model classifier made use of a CNN base with a single-layered FNN connected at the last convolutional layer (Fig 4). The last convolutional layer was flattened to allow connectivity to the FNN layer. The filter size and the filter skip were kept the same for all convolutional layers. Furthermore, the number of kernels in the CNN progressively increased by a factor of $2^N$ from the input layer to the output layer; where the variable $N$ denotes the $N^{th}$ convolutional layer.

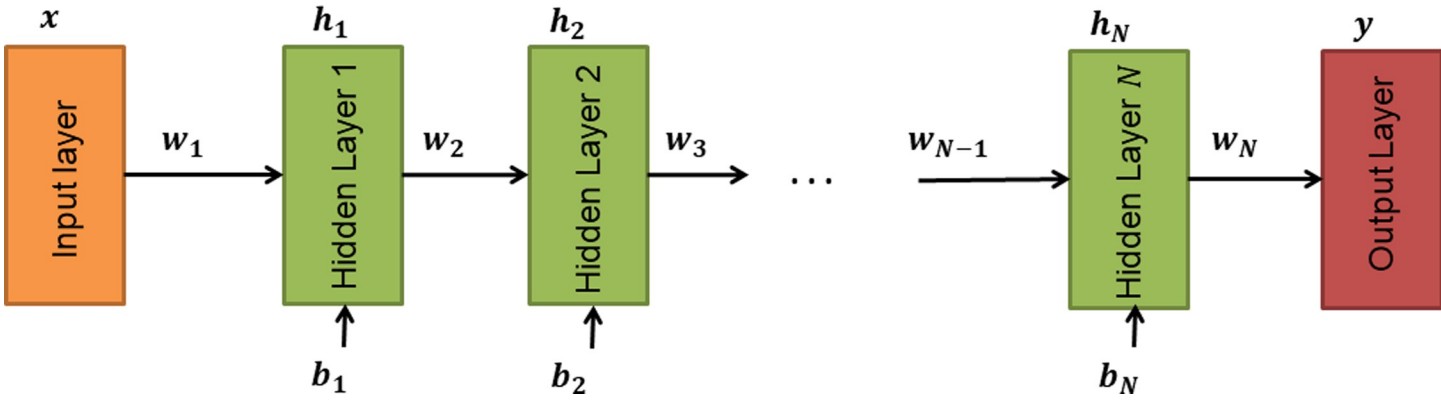

**Fig 2. FNN design architecture.**

### Genetic algorithm design

Each NN was optimized using genetic algorithm (GA). All designed GA had the same hyper-parameter configurations for the maximum number of generations ($G$), the number of individuals ($N$), the mutation rate ($\%m$), the crossover method, and the fitness function. An individual is a solution that is made up of a unique combination of NN parameters. The generation of individuals begins once each parameter has been accurately defined. Individuals are created at random in this method. This stochastic generation of individuals refers to the selection of a random value for each parameter of each individual. The GA architecture per designed NN only varied in terms of individual gene expression; since each NN type had different architectures and hence different NN hyperparameters. To limit the GA search space, the values for each hyperparameter were bounded in the initialization process. Table 1 summarizes the gene expression for each NN.

A population of 30 individuals was created for each designed GA, which was evolved up to 30 generations. Individuals were ranked using the validation set accuracy, where individuals exhibiting higher validation set accuracy were declared fitter individuals. An elitism criterion was implemented where 50% of the fittest individuals were merited 100% chance to mate to a

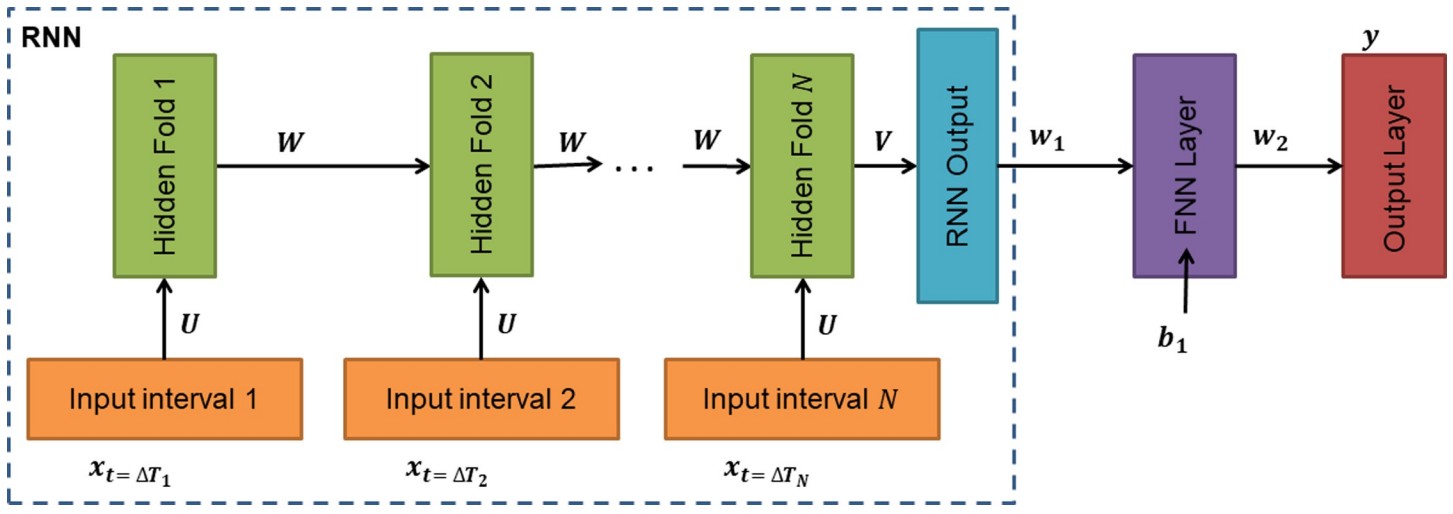

**Fig 3. RNN design architecture.**

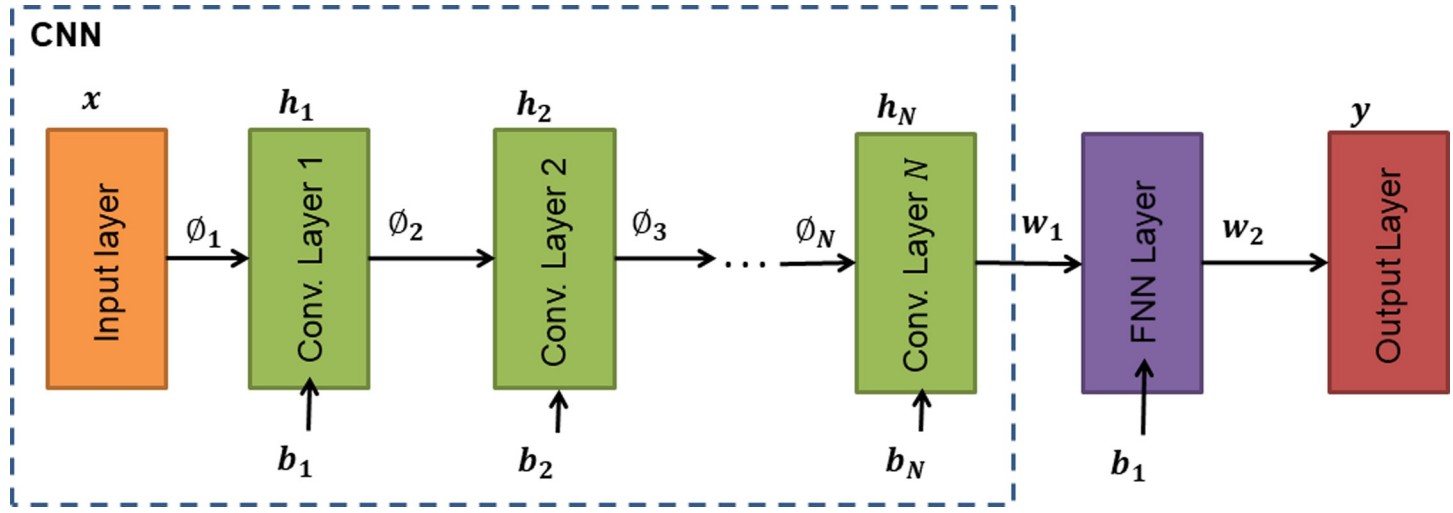

**Fig 4. CNN design architecture.**

respective random individual within the population while the other 50% were replaced by new individuals resulted in the crossover of elite individuals. Elite individuals were carried over to the next generation. Considering mutation, the 50% least fit individuals were assigned a 5% mutation rate where a completely random chromosome was assigned to a mutated individual. The GA was terminated when the number of iterations reached the maximum generation

**Table 1. Neural network hyperparameters for GA.**

| GA-optimized hyperparameters | | | | | |
|---|---|---|---|---|---|
| **FNN (N = 2, N = 4, N = 8)** | | **RNN** | | **CNN** | |
| variable | search space | variable | search space | variable | search space |
| Epoch ($E$) | $[10,300] \in \mathbb{N}$ | Epoch ($E$) | $[10,300] \in \mathbb{N}$ | Epoch ($E$) | $[10,300] \in \mathbb{N}$ |
| Learning rate ($LR$) | $10^{-k} \mid k = [0,7] \in \mathbb{R}$ | Learning rate ($LR$) | $10^{-k} \mid k = [0,7] \in \mathbb{R}$ | Learning rate ($LR$) | $10^{-k} \mid k = [0,7] \in \mathbb{R}$ |
| Neurons per layer ($l$) | $[2,30] \in \mathbb{N}$ | RNN input partitions ($\Delta T$) | $[10,100] \in \mathbb{N}$ | Number of conv. Layers ($N$) | $[1,5] \in \mathbb{N}$ |
| AdaGrad optimization constant ($\varepsilon$) | $10^{-k} \mid k = [0,7] \in \mathbb{R}$ | Neurons per fold ($l_f$) | $[2,30] \in \mathbb{N}$ | Filter size ($N(\emptyset)$) | $[2,11] \in \mathbb{N}$ |
| | | Neurons per FNN layer ($l$) | $[2,30] \in \mathbb{N}$ | Filter skip ($s$) | $[2,N(\emptyset)] \in \mathbb{N}$ |
| | | AdaGrad optimization constant ($\varepsilon$) | $10^{-k} \mid k = [0,7] \in \mathbb{R}$ | Kernel count for N = 1 ($k$) | $2^k \mid k = [1,5] \in \mathbb{N}$ |
| | | | | Neurons per FNN layer ($l$) | $[2,30] \in \mathbb{N}$ |
| | | | | AdaGrad optimization constant ($\varepsilon$) | $10^{-k} \mid k = [0,7] \in \mathbb{R}$ |
| **Non-optimized hyperparameters** | | | | | |
| Weight Initialization | | Gaussian Random: $\sigma = \sqrt{\frac{1}{input\ size}}, \mu = 0$ | | | |
| Input Normalization | | Z-score normalization: $\bar{x} := \frac{\bar{x} - mean(\bar{x})}{std(\bar{x})}$ | | | |
| Activation Functions | | Input/Hidden layers–$SELU$ () | | | |
| | | Output layer–$softmax$ () | | | |
| NN cost function | | Binary cross-entropy | | | |
| FNN dropout/ regularization | | 90% SELU dropout | | | |
| Training optimizer | | AdaGrad stochastic gradient descent (SGD) | | | |

count of 30. The fittest individual in the 30$^{th}$ population is taken as the GA's resulting solution. The GA per NN design was repeated for over 50 trials to ensure stability. All the said hyperparameters of the GA are summarized in Table 2.

## Evaluation of models

Stratified $k$-fold cross-validation was performed to select the best design among the models, where $k = 10$. In the 10-fold cross-validation, the data was divided into $k = 10$ folds of equal sizes consisting of data of approximately equal quantity for each class. 70% of the spectral data set, $X_{SD}$, were used for the training set $S_{TR} \subset X_{SD}$, and the remaining 30% were equally divided for the validation set $S_V \subset X_{SD}$ (15%) and the test set $S_{TS} \subset X_{SD}$ (15%). To ensure greater stability, the cross-validation procedure was repeated over 50 trials ($T$) [50]. The elements of the sets $S_{TR}$, $S_V$, and $S_{TS}$ were randomly reselected from $X_{SD}$ for each trial. The sets satisfied the criteria $S_{TR}^{(i)} \cup S_V^{(i)} \cup S_{TS}^{(i)} = X_{SD}$ and $S_{TR}^{(i)} \cap S_V^{(i)} \cap S_{TS}^{(i)} = \emptyset \forall i \in \mathbb{N} \leq T$. For each set, the ratio of malignant and benign samples was preserved.

To evaluate the performance of each model, the metrics to be quantified were the metrics: area under the curve (AUC), accuracy (ACC), positive predictive value (PPV), negative predictive value (NPV), recall rate (RR), and specificity rate (SR). The overall mean and standard deviation of the metrics over the 50 trials were obtained using the formulas

$$M_m = \frac{1}{T} \sum_1^T \left( \frac{1}{N} \sum_1^N m_{n,t} \right) \tag{5}$$

$$\sum_m = \frac{1}{T} \sum_1^T \left( \sqrt{\frac{\sum_1^N \left( m_{n,t} - \bar{m}_{n,t} \right)^2}{N-1}} \right) \tag{6}$$

in which $M_m$ and $\Sigma_m$ are the overall mean and overall standard deviation of a metric $m$, where $m_{n,t}$ is the metric value of a metric $m$, for a trial $t \in \mathbb{N} \leq T$ and a fold $n \in \mathbb{N} \leq 10$. In addition, the variable $\bar{m}_{n,t}$ in Eq 6 is the $N$-fold mean of a metric $m$, which is also equal to $\frac{1}{N} \sum_1^N m_{n,t}$ from Eq 5.

## Pseudo-clinical test

122 FFPE lung cells or tissue blocks used to acquire the spectral data set were reevaluated by two (2) external evaluators (pathologists) blinded to the original diagnosis of the respective study sites. Microscopic analysis was performed by the external evaluators by identifying H&E-stained sections of the samples as either malignant or benign. After which, the samples

**Table 2. Hyperparameters of GA design.**

| Hyperparameter | Description/Value |
|---|---|
| Maximum number of generations ($G$) | 30 |
| Number of individuals ($N$) | 30 |
| Mutation rate (%$m$) | 0% for the fittest 50%; else 5% |
| Crossover rate (%$c$) | 100% for fittest 50%; else 0% |
| Crossover method | Single point crossover |
| Fitness function | Validation set accuracy |
| Elitism | Fittest 50% as parents with 100% survival rate, and 50% as new individuals |
| Termination criteria | Generation count reaches $G$ |

were clustered (concordant or discordant) based on the results of the reevaluation. To ascertain the probability of discordant lung tissue samples as either malignant or benign in origin, the NN models were used to provide a prediction per sample.

In the prediction process, the NN models were retrained using the concordant lung spectral data as the training set while the discordant spectral data were used as the test set. Each NN type (FNN2, FNN4, FNN8, RNN, and CNN) underwent 50 trials and was run over 10 times to account for the instability of the NN models. The hyperparameters derived from the GA optimization process were used to design the 50 NNs per model type. The average probability presented by each NN model design for each sample was then determined.

# Results

## Spectral data

The spectral data set consisted of 122 spectral vectors comprising 56 malignant FTIR data and 66 benign FTIR data. The data set was further reduced to 118 (53 malignant and 65 benign) by the removal of 4 outlier spectra that had peaks that deviated from the rest of the spectral data. Each spectral vector consisted of 462 elements which constituted absorbance readings within the fingerprint region of 1800 cm$^{-1}$ to 850 cm$^{-1}$ at ~2 cm$^{-1}$ steps. The clinical characteristics of the lung samples were presented in the study of Bangaoil *et al.* [21]. The median absorbance spectrum of the benign and malignant lung tissue samples is shown in Fig 5.

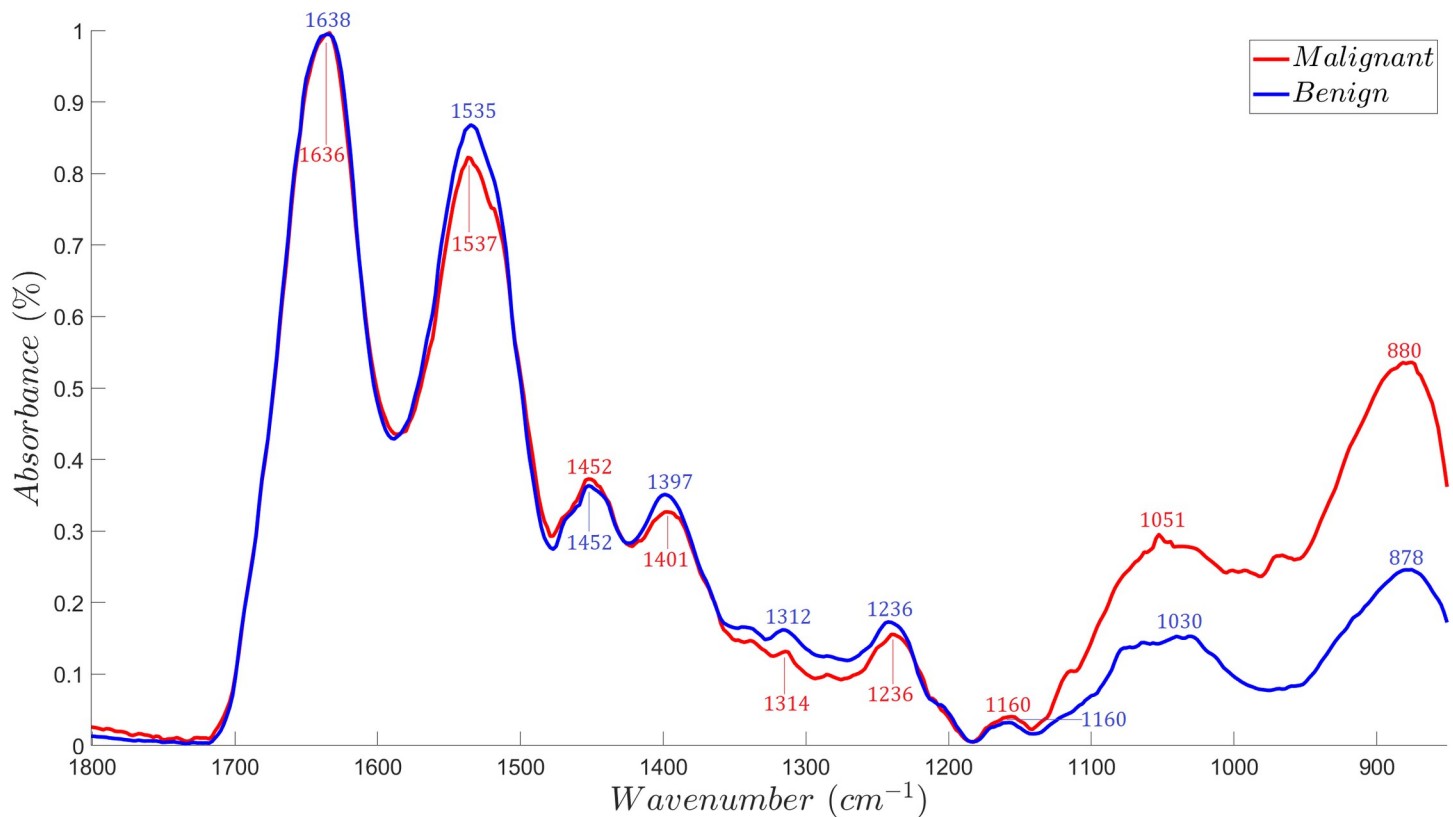

**Fig 5. Median ATR-FTIR absorbance spectra of malignant (*n* = 53) and benign (*n* = 65) lung tissue samples.** The figure shows the median FTIR spectrum of malignant and benign lung tissue samples and their corresponding peaks identified via visual analysis.

## Visual peak analysis of spectral data

All of the visually identified absorbance peaks for malignant and benign samples in the finger-print spectral region, with their corresponding functional group, vibrational mode, molecular source assignments, and $p$-values are indicated in Table 3. The test of homogeneity demonstrated distinct differences ($p<0.05$) between the absorbance peaks of malignant and benign samples, specifically at bands, 1537 cm$^{-1}$ / 1535 cm$^{-1}$, 1314 cm$^{-1}$ / 1312 cm$^{-1}$, 1051 cm$^{-1}$ / 1030 cm$^{-1}$, and 880 cm$^{-1}$ / 878 cm$^{-1}$. No distinct differences ($p>0.05$) were noted at bands 1636 cm$^{-1}$ /1638 cm$^{-1}$, 1452 cm$^{-1}$ / 1452 cm$^{-1}$, 1401 cm$^{-1}$ / 1397 cm$^{-1}$, 1236 cm$^{-1}$ / 1236 cm$^{-1}$, 1160 cm$^{-1}$ / 1160 cm$^{-1}$. This implies that identified molecular sources that corresponded to the wavenumbers with significant differences were distinct among the two classes. Amide II proteins were found to have significantly decreased absorbances in malignant samples. Phosphorylated protein and glycogen, conversely, were found to have increased absorbance values in malignant samples. This finding suggests that identified molecular sources, amide II, glycogen, and phosphorylated protein, may be considered as a point of comparison between malignant and benign classes.

## Principal component analysis of spectral data

The variation between benign and malignant samples is shown in Fig 6. As presented in the PCA plot, the first principal component $F_1$ was associated with 85.65% of the variability, while only 5.67% was associated with the second principal component $F_2$. The majority of the benign samples were situated in the negative domain of the $F_1$ axis while most of the malignant samples were scattered throughout the $F_1$ axis. A clear separation between the two sample classes was not distinct, as shown in the aforementioned figure, hence implying that some malignant and benign samples shared some characteristics and thus difficult to differentiate.

The resulting variance for each biomolecular source in Table 3 was also shown in Fig 6. Phosphorylated protein was associated with the highest variance in the $F_1$ axis and is positively correlated with glycogen.

## Genetic algorithm optimization

The GA fitness over generation curves shown in Fig 7 suggests that the NN hyperparameter tuning via GA for each NN model was able to converge, or at least obtain a steady-state solution of satisfactory performance.

**Table 3. Computation of the spectrum variables (peak positions and normalized absorbances) of malignant and benign lung samples in the fingerprint IR region (1800 cm$^{-1}$ to 850 cm$^{-1}$).**

| Malignant Samples (n = 53) | | Benign Samples (n = 65) | | Functional Group | Vibrational Mode | Molecular Source [43, 51–55] | $p$-value* |
|---|---|---|---|---|---|---|---|
| Peak Position | Mean abs ± SD | Peak Position | Mean abs ± SD | | | | |
| 1636 | 0.9605 ± 0.0856 | 1638 | 0.9885± 0.0139 | O = C–N–H | $v$(CO), $v$(CN) | Amide I, protein | 0.6991 |
| **1537** | **0.7884 ± 0.0898** | **1535** | **0.8585 ± 0.0491** | **O = C–N–H** | **γ(N–H), $v$(C–C), $v$(C–N)** | **Amide II, protein** | **<< 0.05** |
| 1452 | 0.3659 ± 0.0687 | 1452 | 0.3714± 0.0510 | –(CH2)$_n$–(CH3)$_n$– | δ$_{as}$(CH3), δ$_{as}$(CH2), δ$_s$(CH3) | Lipids | 0.9851 |
| 1401 | 0.3244 ± 0.0694 | 1397 | 0.3480 ± 0.0542 | –(CH2)$_n$– | δ$_s$(CH3) | Lipids | 0.0590 |
| **1314** | **0.1334 ± 0.0582** | **1312** | **0.1711 ± 0.0591** | - | - | - | **<< 0.05** |
| 1236 | 0.165 ± 0.0766 | 1236 | 0.1921 ± 0.0775 | RO–PO2$^-$–OR | $v_{as}$(PO2$^-$) | DNA, RNA, phospholipids | 0.1138 |
| 1160 | 0.0467 ± 0.0343 | 1160 | 0.0506 ± 0.0475 | C–O–H | $v$(CO), γ(COH) | Carbohydrates | 0.4302 |
| **1051** | **0.3131 ± 0.1274** | **1030** | **0.1716 ± 0.0875** | **C–O–H** | ***def*(CHO)** | **Glycogen** | **<< 0.05** |
| **880** | **0.5786 ± 0.3124** | **878** | **0.2595 ± 0.1665** | **C–O–P** | **$v$(COP)** | **Phosphorylated protein** | **<< 0.05** |

* Mann-Whitney $U$ test (two-tailed); significant when $p<0.05$.

**Values in bold refer to significantly different peak absorbance between malignant and benign samples ($p>0.05$).

**Abbreviations:** $v$: stretching; δ: bending; γ: wagging, twisting and rocking; $s$: symmetric; $as$: asymmetric; $def$: deformation.

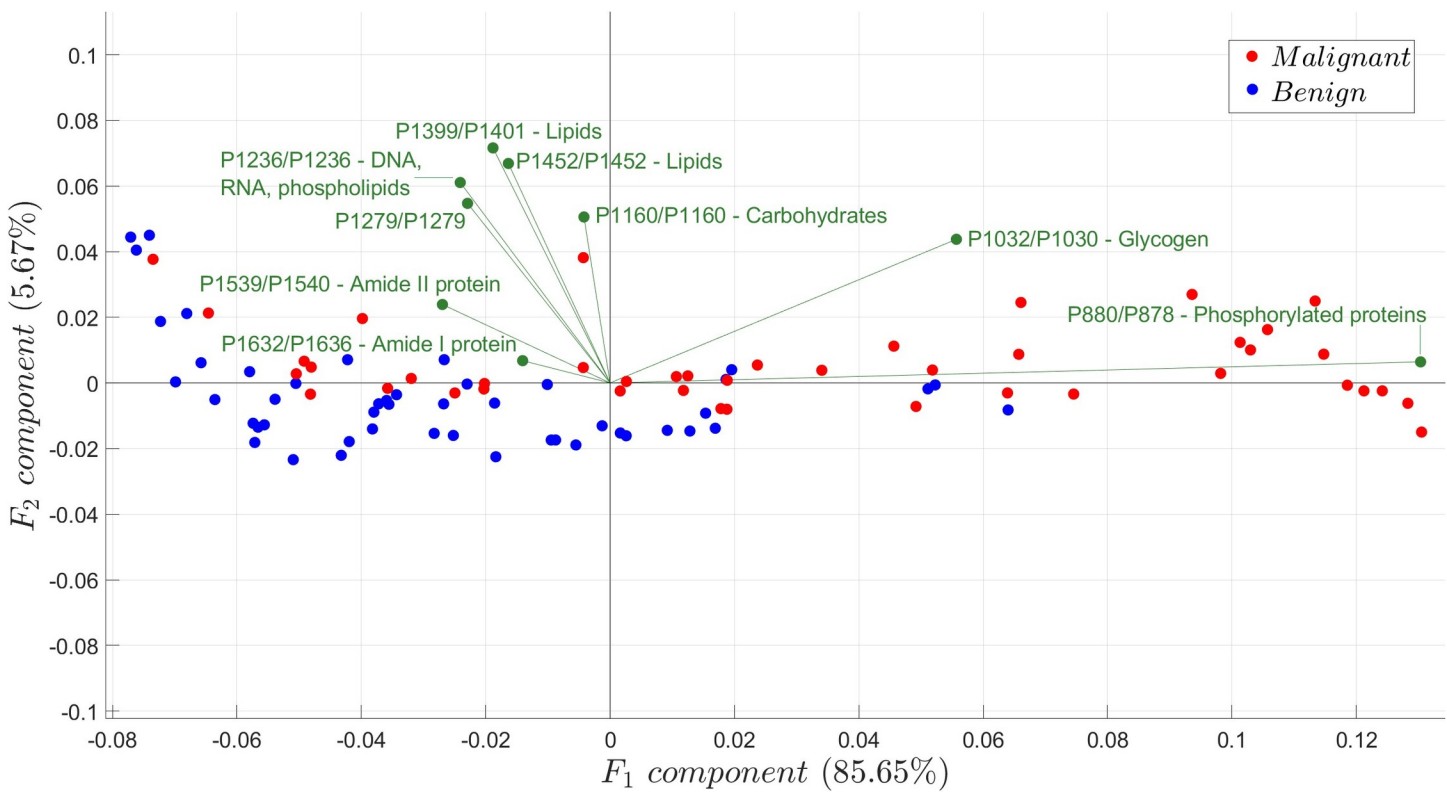

**Fig 6. PCA biplot showing the distribution of malignant and benign samples and the variances contributed by each biomolecule.** The red points represent the malignant samples while the blue points represent the benign samples.

Visually evident is that the GA performed for the FNN models were the fastest to obtain steady state performance, where convergence was evident at about the 9th to 13th generation. The GA for the CNN model, on the other hand, was able to converge at about the 14th generation while the GA for the RNN model was able to converge only at about the 20th or later generations, which took the longest. Also evident from each plot are the comparatively larger range of values observed among RNN-type individuals per generation. Such can also be observed from the earlier generations among CNN-type individuals. The slow convergence of the RNN and the CNN model might be attributed to the larger search space considered for their cases. Relative to the FNN models, which only optimized 3 hyperparameters, the RNN and the CNN models considered 4, and 7 hyperparameters respectively. The larger hyperparameter scope may also explain the high variation of RNN-type and CNN-type individuals during the early generations, while the FNN-type individuals varied relatively less. It is worth noting, however, that the range of validation set accuracy performance among CNN-type individuals decreased in magnitude as the generation progressed, while those of RNN-type individuals were still significantly varied. It might be the case that RNN models of very good performance ($ACC_v > 90\%$) were harder to find and train within the GA-RNN search space relative to the FNNs and the CNN. This might also be the reason why the RNN converged at a lower value of validation accuracy relative to the other models. Hence, there might be a need for the GA to extend longer for the RNN case in order to obtain a higher average performance metric.

From all GAs performed, it can be seen that the average validation set accuracy of the NN models are higher than their training set accuracy. Such may be attributed to the high dropout

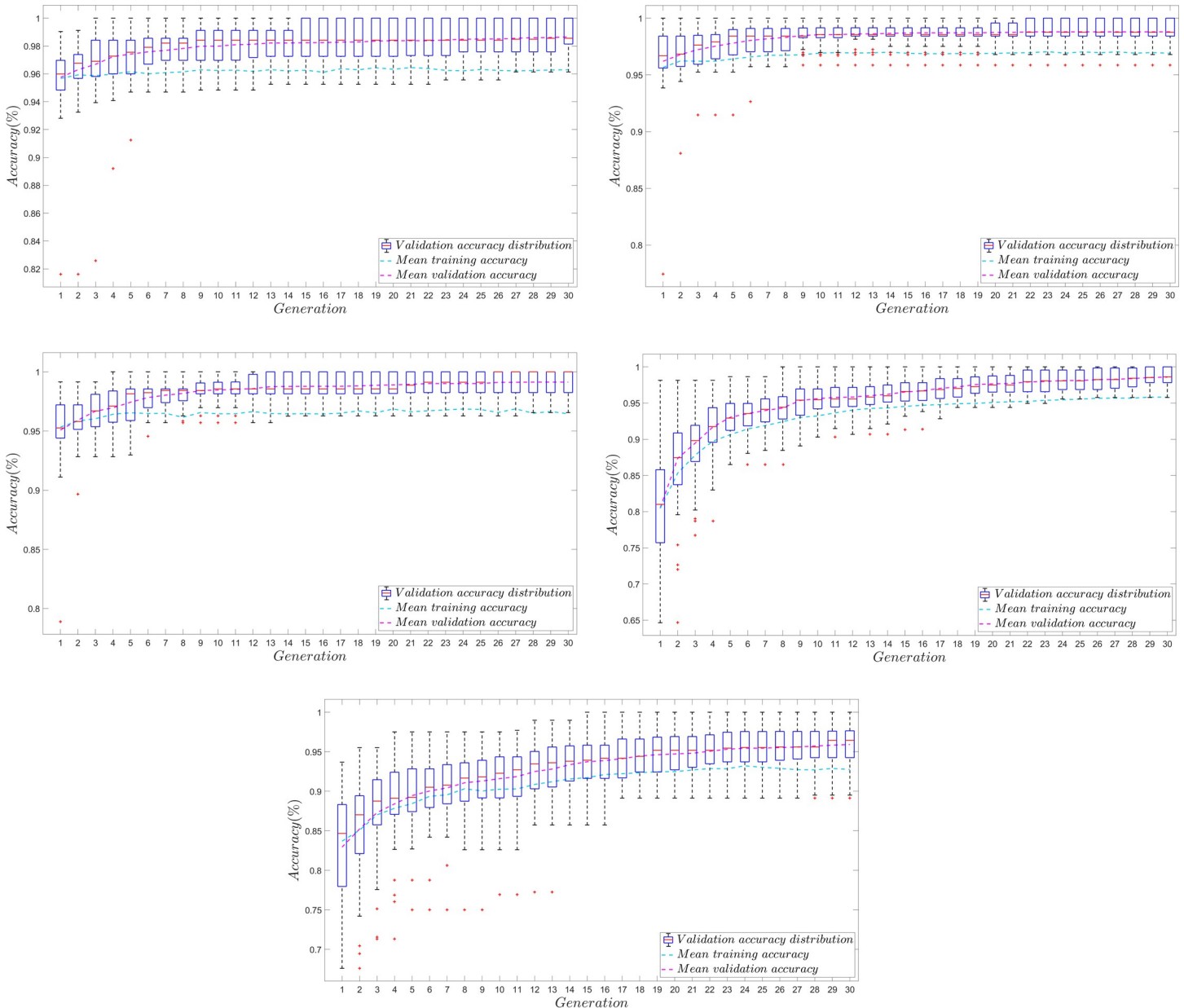

**Fig 7. Average performance accuracy of NN models per generation.** The plots show the average accuracy of each NN model during the GA-based NN hyperparameter tuning process. The averaged metric shown for each generation is derived from the metric of the best individual over 50 trials. Evident in the GA plots, the FNN models were the fastest to achieve steady-state performance while the RNN model was the slowest. The RNN plot also shows a comparatively larger range of values per generation, which may suggest that the search space for RNN models of very high accuracy is relatively smaller than those of the FNNs and the CNN; hence RNN models may be the most difficult to tune. **A.** Average performance accuracy of FNN2-type individuals per generation. **B.** Average performance accuracy of FNN4-type individuals per generation. **C.** Average performance accuracy of FNN8-type individuals per generation. **D.** Average performance accuracy of CNN-type individuals per generation. **E.** Average performance accuracy of RNN-type individuals per generation.

magnitude (*90%*) assumed for each NN. Due to the dropout, it might have been the case that the NNs were limited of computational power to generalize well during training due to reduced number of parameters. However, during validation, since dropout is no longer implemented, the increased number of parameters might have helped it generalize better [56]. The high values for both the training and the validation accuracy for all models suggest that they

are not over-fitted models; and thus poorer metric from the more complex models (those of more parameters such as the CNN and the RNN models) might need more training data and/or training time [57].

## Genetic algorithm-optimized hyperparameters

The hyperparameters of the neural networks are summarized in Tables 4–6. The tabulated median and range value for each hyperparameter quantifies the average value and the search space per hyperparameter which results in an increased probability of attaining excellent performance. The obtained set of median values and ranges per hyperparameter provides a specific search space for future studies concerning NN design using FTIR input data. Through narrowing down the search space, future NN models utilizing FTIR input may be trained faster and more efficiently even for very large data sets; hence providing better predictive capability. For FNNs, both the epoch and the learning rate showed decreasing behavior at increasing network depth. On the other hand, no distinct trend was observed for the number of neurons per layer as a function of network depth, since the trained network's median values were relatively similar. The results, as shown in Table 4, imply that for FNNs, deeper models required less training iterations and less learning rate for them to par shallower models; or at least perform within excellent standards. Results regarding the epoch hyperparameter were consistent with the usual NN design architecture where more complicated models–those usually having a higher number of parameters–tend to be better when trained for sufficiently less iterations to avoid overfitting.

The results concerning FNN learning rates were, however, in contrast to usual trends in design, where a larger learning rate usually comes in hand with less training time [58]. Such a trend is usually evident in deeper models where higher learning rates usually compensate for short training time. Considering the trained networks, the observed contrasting behavior might have been due to increased chances of overfitting for deeper networks and/or for networks of high hyperparameter count when using higher values of learning rate. The inherent similarity between classes in the FTIR data set might have also aided the increased probability of overfitting at higher learning rates since NNs usually tend to overfit when distinguishing very similar classes. Overall, the results show that deeper models, when trained with higher learning rates, may have performed worse during validation and testing due to overfitting patterns from the training set or due to large parameter oscillations.

Table 5 shows the hyperparameters of the RNN model. In comparison to the FNN models, both the RNN median epoch and median learning rate are less than those of FNN8's. This behavior is consistent with the observed FNN trend and its given explanation that the designed RNN architecture had more parameters than the FNN8 architecture. However, it must be noted that the FNN layer of the designed RNN had a greater median neuron count ($n = 21$) than that of FNN2 ($n = 17$). The increase in neurons may imply that the spatial features produced by the RNN design might have been more complex than the input's, making it more difficult to differentiate and thus requiring more computational power. The increased complexity

**Table 4. FNN hyperparameters.**

| | Median (25th percentile, 75th percentile) | | |
|---|---|---|---|
| | **Epoch** | **Learning Rate** | **Neurons per Layer** |
| *FNN2* | 241 (201.5, 268) | $3.95 \times 10^{-3}$ ($1.83 \times 10^{-3}$, $1.59 \times 10^{-2}$) | 17 (7.75, 24) |
| *FNN4* | 229.5 (166, 267) | $2.17 \times 10^{-3}$ ($1.60 \times 10^{-3}$, $5.01 \times 10^{-3}$) | 19 (15.75, 25.5) |
| *FNN8* | 218.5 (189, 270.25) | $1.75 \times 10^{-3}$ ($1.13 \times 10^{-3}$, $3.45 \times 10^{-3}$) | 18 (13, 26) |

**Table 5. RNN hyperparameters.**

| | Epoch | Learning Rate | RNN Neurons per Fold | FNN Neurons per Layer |
|---|---|---|---|---|
| *Median* | 202.5 | $1.43 \times 10^{-3}$ | 24 | 21 |
| *$25^{th}$ Percentile* | 152.5 | $3.77 \times 10^{-5}$ | 21 | 17.75 |
| *$75^{th}$ Percentile* | 264.25 | $8.60 \times 10^{-2}$ | 28.25 | 26.25 |

produced by the RNN layer is also evident from the decreased search space (for top individuals) of the RNN's FNN layer.

Table 6 shows the hyperparameters of the CNN model. Similar to that of the RNN, the CNN model has an epoch and learning rate which are less than that of FNN8's. Implications of its lessened training time and learning rate relative to those of the FNNs follow the same reasoning as that of the RNN. However, as shown in Table 6, the FNN layer of the designed CNN architecture quantifies to about the same neuron count per layer as those of the FNN2. This implies that the designed CNN layer may have been able to translate the data set to a more separable or similarly separable dimension relative to its initial dimension. Compared to the RNN, this shows that the CNN model may have been able to successfully determine more distinguishable patterns for separability; hence may serve as a better or par model to the FNNs. As per the design, most CNN generated by the GA was not able to attain good performance when designed to have deep convolutional layers (N > 2). This may be due to the fact that deeper CNNs may have produced more complex abstraction that may be difficult for the FNN layer to distinguish; hence may have necessitated either more neurons per layer, longer training, or other modifications in hyperparameters.

## Diagnostic performance of machine learning models

The results of the diagnostic performance metrics of all ML models are summarized in Table 7, giving a side-by-side comparison between the NN models and the classical ML models (DT, RF, NB, LDA, LR, SVM). In general, the metric values of NNs across all performance metrics were notably higher than that of classical models, except for the SVM model whose results were comparable to NNs and was identified as the best benchmark model. Among the NN models, CNN had the highest accuracy (98.45% ± 1.72%), PPV (96.62% ± 2.30%), and SR (96.01% ± 3.09%). On the other hand, RNN had the highest results for NPV (94.03% ± 9.26%) and RR (92.87% ± 13.18%), while FNN4 had the best result for AUC (92.85% ± 9.98%).

In comparison to the best benchmark model, none of the NN models was able to outperform SVM in the AUC metric (99.38% ± 1.97%) since all NNs showed significantly lower AUC (Table 8). However, all NN models surpassed the diagnostic accuracy of SVM by an average of 4.06% for FNN2, 4.27% for FNN4, 3.94% for FNN8, 1.92% for RNN, and 4.18% for CNN. In terms of NPV and SR, the majority of NN models did not significantly differ from SVM, while most of the NNs showed significantly higher PPV.

**Table 6. CNN hyperparameters.**

| | Epoch | Learning Rate | Filter Size | Skip Length | Number of Conv. Layers (N) | Kernel size at N = 1 | FNN neurons per layer |
|---|---|---|---|---|---|---|---|
| *Median* | 205 | $1.07 \times 10^{-3}$ | 8 | 4 | 2 | 8 | 17.5 |
| *$25^{th}$ Percentile* | 162 | $1.04 \times 10^{-5}$ | 6 | 2.75 | 1 | 7 | 14.75 |
| *$75^{th}$ Percentile* | 221.25 | $2.02 \times 10^{-2}$ | 9 | 4 | 2 | 16 | 22 |

**Table 7. Mean and standard deviation of diagnostic performance of all the machine learning models.**

|  | FNN 2 | FNN 4 | FNN 8 | RNN | CNN | DT | RF | NB | LDA | LR | SVM |
|---|---|---|---|---|---|---|---|---|---|---|---|
| *AUC (%)* | 92.41 ± 10.28 | 92.85 ± 9.98 | 92.77 ± 9.62 | 90.40 ± 11.62 | 92.28 ± 7.36 | 78.93 ± 19.87 | 92.15 ± 13.79 | 77.91 ± 21.22 | 62.92 ± 13.44 | 82.16 ± 19.84 | 99.38 ± 1.97 |
| *ACC (%)* | 98.41 ± 4.07 | 98.39 ± 3.57 | 97.61 ± 4.91 | 95.98 ± 6.25 | 98.45 ± 1.72 | 77.58 ± 16.91 | 85.87 ± 15.11 | 75.13 ± 19.14 | 65.45 ± 14.60 | 72.19 ± 18.68 | 94.38 ± 9.69 |
| *PPV (%)* | 94.92 ± 9.49 | 96.03 ± 8.63 | 96.48 ± 8.01 | 90.91 ± 11.99 | 96.62 ± 2.30 | 73.95 ± 30.28 | 83.55 ± 26.14 | 64.65 ± 33.42 | 59.54± 20.84 | 65.25 ± 36.31 | 93.85 ± 14.92 |
| *NPV (%)* | 92.79 ± 11.95 | 93.57 ± 11.28 | 93.22 ± 10.75 | 94.03 ± 9.26 | 90.50 ± 11.92 | 80.11 ± 21.98 | 87.60 ± 18.36 | 82.21 ± 22.01 | 70.25 ± 15.13 | 94.89 ± 30.57 | 94.57 ± 12.11 |
| *SR (%)* | 94.60 ± 11.41 | 94.67 ± 11.57 | 95.30 ± 10.64 | 86.63 ± 17.58 | 96.01 ± 3.09 | 84.36 ± 18.04 | 90.78 ± 14.34 | 79.79 ± 18.68 | 69.08 ± 14.50 | 74.97 ± 25.98 | 96.75 ± 8.03 |
| *RR (%)* | 89.84 ± 16.36 | 90.88 ± 16.13 | 90.10 ± 15.66 | 92.87 ± 13.18 | 89.21 ± 12.93 | 73.35 ± 28.25 | 83.74 ± 24.06 | 71.10 ± 33.83 | 62.13± 17.14 | 64.49 ± 35.74 | 94.46 ± 12.64 |

## Pseudo-clinical test

Among the specimens, 94 samples (42 malignant; 52 benign) were in diagnostic concordance with the external evaluators and the original diagnosis of the study site; while for the 24 samples, the pathologists had discordant readings. The distribution of discordant samples in the PCA score plot is shown in Fig 8. The majority of the discordant benign and malignant samples were situated similarly within the region of the concordant benign and malignant samples, respectively. Moreover, the discordant readings followed a similar distribution, indicating that they were not outliers. The discrimination of the NN models for all discordant samples was consistent with the diagnoses of the respective study sites.

Fig 9 shows the predictive capacity of each NN model for each discordant sample. The median score for each sample corresponds to the model's most probable predictive score, while the range corresponds to the variation of predictive scores for each model per sample. A short measure of range and a high predictive median score imply that the sample is confidently predicted. This infers that the sample's spectrum might be common on the training set; hence, might be a common type of lung tumor. On the other hand, a wide-ranged, relatively low predictive median score implies that a sample differs in spectrum relative to those used in the training set. This infers that the sample may be morphologically and/or chemically different, which may imply that it could be a relatively rarer type of tumor. This may also mean that the sample might have been extracted between the border of malignant and benign tissues. In any case, further investigation is recommended for samples predicted with such characteristic such as a re-examination of its corresponding H&E sample.

In terms of confidence, the FNN models were the most confident in predicting malignancy and benignancy, having median prediction scores that were 80% and above; in which most were well above 99%. The CNN model was also able to obtain high prediction confidence for all samples with the exception of sample NMLB056, in which it was only ~60% confident. Lastly, the median prediction scores for the RNN model were relatively less and were more spread out compared to the other models. Thus, the RNN model may not be reliable, considering the limited data used. After a visual examination of the results shown in Fig 9, the FNN4

**Table 8. Difference of average performance metric between NN models and the SVM model.**

| | Difference of average performance metric (*p*-value)* | | | | | |
|---|---|---|---|---|---|---|
| | **AUC(%)** | **ACC(%)** | **PPV(%)** | **NPV(%)** | **SR(%)** | **RR(%)** |
| *FNN2* | **-6.97** ($\ll$ **0.05)** | **4.03** ($\ll$ **0.05)** | 1.07 (0.1094) | **-1.78 (0.0448)** | **-2.15 (0.0041)** | **-4.62** ($\ll$ **0.05)** |
| *FNN4* | **-6.53** ($\ll$ **0.05)** | **4.01** ($\ll$ **0.05)** | **2.18 (0.0024)** | -1.01 (0.4156) | -2.08 (0.0734) | **-3.57** ($\ll$ **0.05)** |
| *FNN8* | **-6.60** ($\ll$ **0.05)** | **3.23** ($\ll$ **0.05)** | **2.63** ($\ll$ **0.05)** | -1.36 (0.9560) | -1.45 (0.1992) | **-4.36 (0.0185)** |
| *RNN* | **-8.98** ($\ll$ **0.05)** | **1.60 (0.0023)** | **-2.94 (0.0099)** | -0.54 (0.8857) | **-10.12** ($\ll$ **0.05)** | -1.59 (0.0713) |
| *CNN* | **-7.10** ($\ll$ **0.05)** | **4.07** ($\ll$ **0.05)** | **2.77 (0.0015)** | **-4.07** ($\ll$ **0.05)** | -0.74 (0.8493) | **-5.25** ($\ll$ **0.05)** |

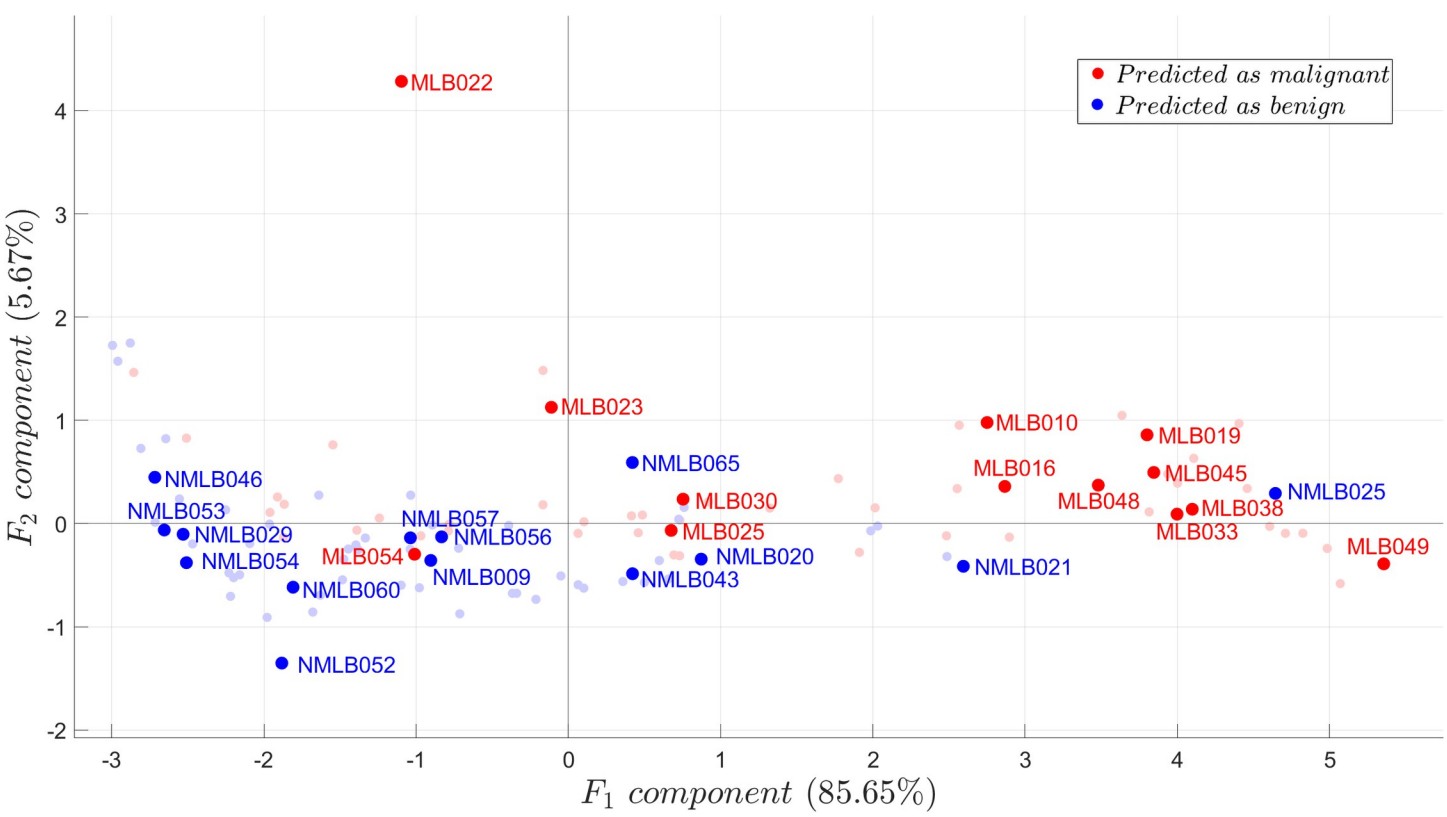

**Fig 8. Distribution of discordant samples from concordant samples via PCA.** The plot shows the diagnosis of the models per discordant samples (dark-colored points) over the distribution of concordant malignant samples (light red) and concordant benign (light blue) samples. The diagnoses of the models for all discordant samples were consistent with the original diagnoses of the study sites.

model was the most suitable model considering the limited training data of the study, since it constituted the model of the highest predictive median score and smallest range for each discordant sample. This implies that the quantity of parameters from FNN4 is suitable to account for the amount of training data ($n = 94$), the number of spectral variables considered ($N(X_{SD})$ = 462), and the variations of spectral data. It is expected, however, that in order to account for a higher variation and more numerous tumor spectrum, a deeper FNN model or a different deeper architecture (such as the RNN and the CNN) may become more appropriate. Regardless of the model, it is very important that multiple instances of NNs be considered for predicting any test sample. This consideration accounts for the instability of solutions derived during the NN training process since the optimization space for NN parameters is non-convex; hence, solutions may be caught up in different local optima [50]. This need for multiple runs is apparent from Fig 9, especially for deeper models in which prediction probabilities range in multiple values.

Depending on the architecture, each NN model showed different degrees of predictive confidence for each sample, as shown in Fig 9. These differences show how different architectures learn the feature variations considered from the training set (set of concordant readings). However, regardless of the model, some discordant samples, in particular NMLB046 from the benign group and MLB054 from the malignant group, were not confidently predicted. This common inference from different NN architecture suggests that these samples significantly vary in spectrum from those in the training set. Further investigation of the causes of difficulty in diagnosing these samples is outside the scope of the current study.

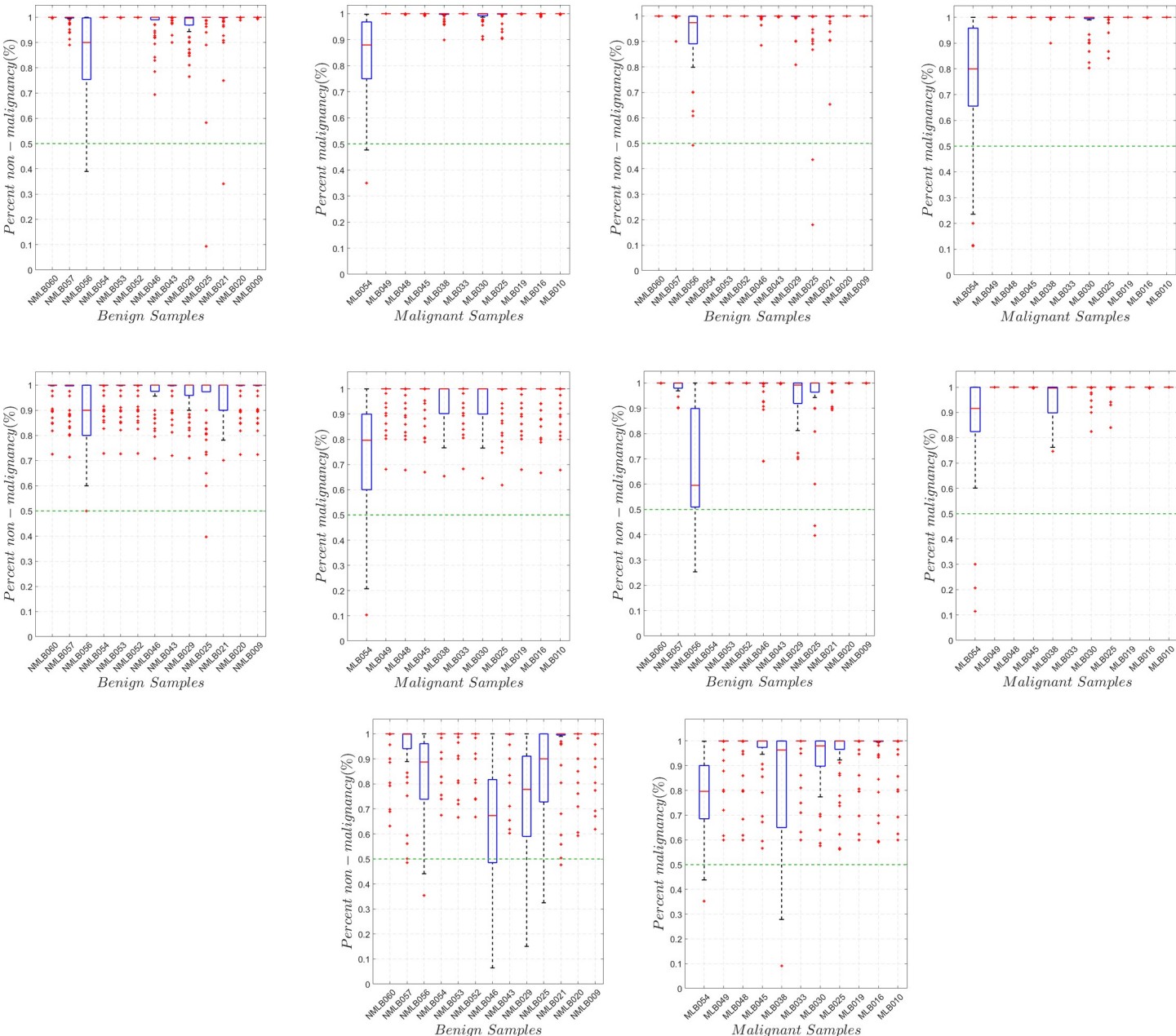

**Fig 9. Prediction probability of NN models per discordant samples.** The figures show the prediction probability of each NN model for the discordant benign (n = 13) and discordant malignant (n = 11) samples. The discordant samples were grouped according to the diagnosis by the pathologist of their respective study sites. All NN models show a median prediction score that is above the 0.5 (50%) mark, meaning that all the models had the same prediction as that of the diagnosis of the pathologist. **A.** Prediction probability of FNN2 models per discordant samples. **B.** Prediction probability of FNN4 models per discordant samples. **C.** Prediction probability of FNN8 models per discordant samples. **D.** Prediction probability of CNN models per discordant samples. **E.** Prediction probability of RNN models per discordant samples.

## Discussion

The potential of artificial neural networks in the detection of lung cancer based on infrared spectroscopy is presented in this study. This method could become an adjunct tool whenever discordant readings among pathologists arise or when there are uncertainties in the malignancy of the H&E-stained specimens. FTIR spectroscopy is known to be an assistive diagnostic

technique for multiple cancers as it yields highly specific and accurate results [59]. Different chemometric techniques with the likes of artificial neural networks (ANN) and principal component analysis (PCA) have been utilized to bring about chemo-physical evidence from spectral data [60]. With the surge of development in computer-aided diagnosis, most medical imaging techniques, such as mammography, chest radiographs, and chest CT, utilized for diagnosis have been focused on and are growing exponentially [61]. However, even with these advancements, detection and classification remain difficult at some point [62]. Thus, automated FTIR spectroscopy, albeit less intuitive than medical imaging, now receives more attention as it could reduce intra- and inter-operator variability [63]. Its application with neural networks may be seen in the study of Santillan *et al.*, where infrared spectral data of thyroid tumors were discriminated by NNs. This study produced a 98% accuracy using its RNN model and even outperformed the LDA model in most metrics [22]. Not only does it allow a standardized method for diagnosis, but it also permits automatic data analysis of large data sets by algorithms done by non-spectroscopists [63].

To effectively understand the differences in biochemical composition of the lung samples, their clinical characteristics must be taken into consideration. The majority of malignant lung samples used in this study were diagnosed as non-small-cell lung carcinoma (NSCLC), mostly in their advanced metastatic form (stage III/IV), and with lung and pleural tissues as main biopsy sources [21, 64]; hence, a more progressive metabolic reprogramming may be observed.

The nine (9) distinct band positions identified by ATR-FTIR spectroscopic analysis elucidated the metabolic changes in malignant samples. Several band positions had shown no significant difference between malignant and benign samples (Table 3). This implies that the spectral data of both classes were difficult to differentiate, especially when using simple linear models. However, there were three (3) band positions that showed significant differences in absorbances.

Band absorbance of 1537 cm$^{-1}$ is within the amide II protein region. There was a significant decrease in amide II proteins in malignant samples compared to benign samples. The decrease in absorbance in the amide band region can be due to proteolysis in the lung cells induced by tumor metabolism [65], resulting in protein remodeling [66, 67].

The peak absorbances for glycogen (1051 cm$^{-1}$) and phosphorylated proteins (880 cm$^{-1}$) were significantly increased in malignant samples. These are aligned with the findings of several lung cancer studies [43, 51, 68, 69], but contrary to the findings of Kaznowska *et al.* where glycogen is decreased in squamous cell carcinoma and adenocarcinoma [70]. Glycogen metabolism plays an important role in cancer cell survival. Guler *et al.*, detected higher amounts of glycogen in cancer cells compared to normal cells which suggested that its accumulation is one of the essential strategies employed by cancer cells for continuous glucose utilization and metabolic adaptation caused by hypoxia [71–73]. Thus, an increased glycogen level may be considered as a biomarker for lung cancer, suggesting energy metabolism during synthesis, cell proliferation, and cell survival [52, 73]. Similar to the findings of Bangaoil *et al.*, there was no significant difference found in the absorbance intensity of carbohydrates (1160 cm$^{-1}$) between the malignant and benign samples [21]. It is likely that both malignant and benign samples utilize glycogen, hence no significant difference in absorption intensity of carbohydrates was observed.

On the other hand, an increase in absorbance peak in the phosphorylated protein region was apparent, exhibiting hyperphosphorylation in malignant cells [74]. Lee *et al.* also found a greater occurrence of phosphorylation in lung malignant cells compared to normal lung cells [75]. Cell deregulation, such as phosphorylation modification of p53, a tumor suppressor protein, may have contributed to the increase in absorbance in the phosphorylated protein region [75].

It can be noted that while the current study and that of Bangaoil *et al.* analyzed the same spectral datasets, the latter observed no significant difference in the phosphorylated protein

(880 cm$^{-1}$/ 878 cm$^{-1}$) region of malignant and benign lung cancer samples [21]. Compared to Bangaoil *et al.*, normalization was done in this current study, which is a crucial step to remove any biases and discrepancies in absorbance readings due to confounding factors such as varying densities of samples [52, 76]. If a non-normalized spectral dataset was used, the discrimination done by the machine learning models would primarily be based on the absorbance intensity, whereas the use of normalized spectra would highlight the distinct biochemical structures that could differentiate malignant from benign samples [76].

Using normalized ATR-FTIR data, the NNs exhibited an accuracy of at least 95%. This metric proves the superior overall effectiveness of NNs in discriminating malignant from benign lung spectral data. Among the NNs, CNN was the most accurate in detecting true malignant spectra, while RNN was the most accurate in detecting benign spectra. Overall, CNN had the best diagnostic performance. This may be explained by the fact that CNN is known to provide exemplary results in the reduction of parameters [77]. Moreover, Acquarelli *et al.* demonstrated that even a shallow CNN achieved significantly better classification accuracy using one-dimensional data (e.g. vibrational spectroscopic data) compared to other machine learning models such as LDA and k-nearest neighbors [78].

Among the classical models, only SVM had performance metrics comparable to that of NNs. The significantly higher ACC rate of the NNs compared to SVM denotes that NNs are generally better overall classifiers. ACC is a viable intuitive metric of comparison since the spectral dataset used in the study was roughly balanced (53 malignant and 65 benign), and both classes (*i.e.*, malignant and benign) are considered of equal importance. However, such metric does not entail better individual class effectiveness [79]. The non-significantly different SR values of most NNs and SVM indicate the equal effectiveness of the models in identifying benign spectra. Meanwhile, the RR values of NNs were significantly lower than that of SVM, implying that the latter is more effective in identifying malignant spectra.

In terms of predictive values, there were no significant differences in the NPVs of NNs and SVM, signifying equal predictive power in identifying truly benign samples. Conversely, the majority of NNs were observed to have significantly higher PPVs compared to SVM. Such denotes that most NNs were better classifiers in terms of identification of malignant spectra as truly malignant. A high PPV reduces the occurrence of false positives [80]. Falsely classifying patients as having malignant tumors can result in negative short-term psychosocial consequences [81], possibly resulting in non-adherence to subsequent lung cancer screening tests [82]. Moreover, misdiagnosing patients with rare benign tumors that mimic malignant neoplasm will not only entail higher medical care costs [83], but will also lead to an intractable condition due to further surgical procedures, chemotherapy, and radiotherapy that are designed for treating malignant cases [84].

The choice between NNs and SVM is highly dependent on the point of interest presented by the problem domain. If there is a need for more accurate true malignant infrared spectra detection, NNs, specifically CNN, may be deemed more superior than SVM. Conversely, for a more accurate true benign infrared spectra detection, either NNs or SVM may be utilized since both were observed to be comparable.

Newman-Toker *et al.* and Richards *et al.* stated that lung cancer recorded the highest rate of variation in error and harm frequency that even the screening tests available are below the recommended levels [85, 86]. A missed lung cancer diagnosis is a common medicolegal issue despite the availability of advanced imaging techniques for diagnosis. Chest radiography amounts to 90% of misdiagnosis, while the remaining 10% are from CT examinations and other imaging studies [87]. Assessing chest films and scans is subjective and observations may vary from one radiologist to the other [88], hence misdiagnosis can occur. While observer error remains as one of the biggest factors for misdiagnosis [89], other pulmonary diseases,

such as sarcoidosis [90] and tuberculosis especially in countries with limited facilities [91], may also play role in misleading the diagnosis. Bangaoil *et al.* further mentioned that the benign samples that were subclustered, through hierarchical component analysis (HCA), with malignant samples came from patients who suffered from pulmonary tuberculosis and chronic granulomatous disease [21]. Earlier studies have shown that these two diseases may potentially lead to the development of lung carcinoma [69, 72]. This entails that appropriate timing to a certain stage is also vital. Missed lung cancer diagnosis also has an impact on the prognosis of the disease–an earlier diagnosis will constitute a better prognosis and a wide possibility for eventual treatment [89].

It is then undeniably beneficial to establish an exceptional and reliable lung cancer diagnostic process. This part of the medical process is heavily reliant on the knowledge, skills, and experience of the pathologist and the reliability of the samples gathered [84]. There is a potential for the improvement of lung cancer diagnosis by introducing an automated method of classifying tissues based on the vibrational spectroscopic patterns in the infrared fingerprint region. A similar study done by Kaznowska *et al.* showed that FTIR spectral data from distinct regions can be used to discriminate between benign and malignant lung cancer tissue using PCA-LDA and a physics-based computation model to analyze their data. Their results showed that the model is 78% to 99% sensitive and 65% to 99% specific [70]. Another study by Abbas *et al.* differentiates malignant pleural mesothelioma (MPM) from lung cancer as well as benign pleural effusion (BPE) using the FTIR spectral data from pleural samples wherein hierarchal cluster analysis (HCA) and PCA was applied. Their model was 100% sensitive and specific in differentiating MPM from lung cancer, and 100% sensitive and 88% specific in differentiating MPM from BPE (n = 69; BPE n = 25; LC n = 20; MPM n = 24) [92]. To further advance these findings, more sophisticated computational tools such as NNs may be used in the discrimination of spectral data. The designed NNs of this study are 89.84% to 92.87% sensitive (RR) and 86.63% to 96.01% specific (SR). While the sensitivity and specificity of Kaznowska *et al.*'s study can reach 99% [70], it has higher variability in discriminating samples compared to the NNs. Thus, SVM or NN-aided diagnosis using infrared spectroscopy may be utilized and is a better adjunct tool in identifying truly benign or malignant tissue samples.

Furthermore, the consistency of the diagnoses of the models with the original diagnoses of the respective hospital sites for all discordant samples emphasizes the potential of NNs to provide a second opinion to pathologists in identifying lung cancer. In the clinical setting, the final histological diagnosis, especially in problematic cases, is usually a consensus approach among pathologists. Hence, using infrared spectral data of biopsy tissues may assist in making an accurate diagnosis of the patient.

Albeit the accuracy of the results was exemplary, it is still recommended to apply the NN models in a diagnostic cohort study to test its reliance in the clinical setting and real-world practice. This study utilized a diagnostic case-control research design which creates spectrum biases due to limited samples, ergo the diagnostic performance may be inflated [93, 94]. The obtained performance metrics of the NN from a diagnostic case-control would not suffice as it cannot be applied in the clinical setting [93]. A diagnostic cohort study, on the other hand, is done in a setting similar to that of the real-world practice [95]. Moreover, cohort studies can determine the prevalence of the disease and assess the predictive values of the tests [95]. Diagnostic case controls are commonly utilized in initial studies as it is easier to manage experimental conditions and is exploratory in nature. Diagnostic cohort studies, meanwhile are confirmatory in nature, making the latter more applicable in succeeding studies [95]. Lastly, utilizing biofluids, such as pleural fluids or blood samples, may further expand the use of automated ATR-FTIR in lung cancer early screening process.

## Conclusion

The findings of the study further proved the potential of integrating NN models as a computational tool in diagnosing lung cancer based on ATR-FTIR spectra. The principal component analysis of the spectral data of malignant and benign samples showed evident intersection, and thus simple models such as linear models may show subpar diagnostic performance. NNs generally showed better diagnostic performances compared to other common machine learning models, except SVM which exhibited results that were at par with NNs. Among the NN models, CNN demonstrated the best diagnostic effectiveness (98.45% ± 1.72%) and positive predictive value (96.62% ± 2.30%) while RNN obtained the best negative predictive value (94.03% ± 9.26%).

## Acknowledgments

The authors would like to thank UST Research Center for the Natural and Applied Sciences, Manila, Philippines.

## Author Contributions

**Conceptualization:** Pia Marie Albano, Rock Christian Tomas.

**Data curation:** Eiron John Lugtu, Denise Bernadette Ramos, Alliah Jen Agpalza, Erika Antoinette Cabral, Rian Paolo Carandang, Jennica Elia Dee, Angelica Martinez.

**Formal analysis:** Eiron John Lugtu, Denise Bernadette Ramos, Alliah Jen Agpalza, Erika Antoinette Cabral, Rian Paolo Carandang, Jennica Elia Dee, Angelica Martinez.

**Investigation:** Eiron John Lugtu, Denise Bernadette Ramos, Alliah Jen Agpalza, Erika Antoinette Cabral, Rian Paolo Carandang, Jennica Elia Dee, Angelica Martinez.

**Methodology:** Rock Christian Tomas.

**Project administration:** Pia Marie Albano, Rock Christian Tomas.

**Resources:** Abegail Santillan, Ruth Bangaoil.

**Software:** Rock Christian Tomas.

**Supervision:** Pia Marie Albano, Rock Christian Tomas.

**Validation:** Rian Paolo Carandang, Pia Marie Albano, Rock Christian Tomas.

**Visualization:** Eiron John Lugtu, Denise Bernadette Ramos, Alliah Jen Agpalza, Erika Antoinette Cabral, Jennica Elia Dee, Angelica Martinez, Rock Christian Tomas.

**Writing – original draft:** Eiron John Lugtu, Denise Bernadette Ramos, Alliah Jen Agpalza, Erika Antoinette Cabral, Rian Paolo Carandang, Jennica Elia Dee, Angelica Martinez.

**Writing – review & editing:** Julius Eleazar Jose, Abegail Santillan, Ruth Bangaoil, Pia Marie Albano, Rock Christian Tomas.

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
