## [Decision Letter · Decision Letter 0]

10 Sep 2021

PONE-D-21-23770Artificial neural network in the discrimination of lung cancer based on infrared spectroscopyPLOS ONE

Dear Dr. Lugtu,

Thank you for submitting your manuscript to PLOS ONE. After careful consideration, we feel that it has merit but does not fully meet PLOS ONE’s publication criteria as it currently stands. Therefore, we invite you to submit a revised version of the manuscript that addresses the points raised during the review process.

The reviewers offer a number of technical critiques that I believe are best handled through a major revision. In particular, one point that both reviewers agree on is the viability of the normalization methods used. This caught my eye as well, and I recommend that the authors consider more traditional normalization methods (such as peak normalization) or justify their use of the proposed methods described in Equation 1. In addition, Reviewer 2 brings up some more practical experimental questions that may require significant changes in data analysis that will require additional review.

We look forward to receiving your revised manuscript.

Kind regards,

David Mayerich

Academic Editor

PLOS ONE

Journal Requirements:

Reviewers' comments:

Reviewer's Responses to Questions

**Comments to the Author**

1. Is the manuscript technically sound, and do the data support the conclusions?

Reviewer #1: Yes

Reviewer #2: Yes

2. Has the statistical analysis been performed appropriately and rigorously? 

Reviewer #1: Yes

Reviewer #2: Yes

3. Have the authors made all data underlying the findings in their manuscript fully available?

Reviewer #1: Yes

Reviewer #2: No

4. Is the manuscript presented in an intelligible fashion and written in standard English?

Reviewer #1: Yes

Reviewer #2: Yes

5. Review Comments to the Author

Reviewer #1: This paper is aimed to develope neural network (NN) models to discriminate malignant lung cancer samples from benign ones. 5 different types of NN models have been developed and the performance of these NNs were compared to the performances of 6 different artificial intelligence (AI) methods. CNNs were found to perform best and the nearest AI method was SVM. I have the following comments to improve the manuscript:

Comment 1 : Preprocessing of spectral data: The preprocessing was done by rubber-band baseline correction followed by mean centering (to 0) and std normalization (to 1) using Eq.1. However, these two steps will not be sufficient to make the spectral data to be independent from the equipment, operator, or time of acquisition. (To make this more clear consider an exaggerated case for scaling operation: Assume 100 samples of spectra 90 of which have almost the same appearance but 10 spectra are taken with a gain factor of about 0.1. Now, if the method in the manuscript is used, after preprocessing, the 10 low-amplitude spectra will have high negative values because mean spectra are dominated by high amplitude x-variables. The score values in the PCA plot for these low amplitude spectra will be always highly negative along the pricipal component F1. This may not have been the case. Multiplying, for this case, the low amplitude samples by 10 may have resulted by in positive values for the first principal component.) I would suggest either unit vector normalization or peak normalization to the baseline corrected spectra before applying the scaling in Eq.1.

Comment 2. What were the inputs to the SVM classifier? (First two principal components of the PCA scores, or more?) To detrmine the SVM classifier performance would full-cross validation (Leave -one-out) be more preferrable or reliable?

Comment 3.Several papers appeared in literature recently about the use of infrared spectroscopy in lung cancer characterization and differential diagnosis using different sample types obtained from human individuals. For example see Abbas et al. 2018, J. Biomedical Optics and Kaznowska et al., 2018, Talanta. Brief discussion and comparison are necessary between the results of the present and previously published studies.

Reviewer #2: A research about the application of machine learning combined with ATR-FTIR in the lung cancer diagnosis is described in the manuscript. The experiments and data processing were comprehensively designed and conducted, the description is detailed and the statistics of data is basically appropriate.

Although it is Not a highly innovative idea, this research and the article did give a good practice on presenting the potential of NN models and even other ML methods as an effective tool in diagnosing lung cancer based on ATR-FTIR spectra. Thus, this manuscript could be accepted after a major revision.

The following issues need to be further considered and explained.

1. In ‘Introduction’ section

The evaluation of radiography methods compared to the FTIR method in line 26 is not completely reasonable and just. The FTIR method presented in this article is based on invasive sampling operation, whether it is suitable for routine health check rather than the non- invasive LDCT method? Besides, the limited and fully evaluated risk of exposure to radiation and the issue of the time and the workload demand could not be accounted as the main disadvantages of LDCT test. Please reconsider that is this part of statement fully appropriate for publication.

2. In ‘Materials and methods, Study population and sample preparation’ section

Although the specimen processing and FTIR analysis have been reported in previous publication, it would be helpful for understanding this research and could also make it easier to read if these details were described in this manuscript, including issues such as the size of the samples, the sampling operation and the pretreatment of the specimens before the FTIR analysis, and whether the FTIR spectra were obtained from a small sampling point or as an average result of the entire specimen. Besides, the optical images of the specimens, graphical illustration of the FTIR analyzing process and the FTIR spectra were preferred in the manuscript or at least in Supplementary materials.

3. In ‘Data measure/instrumentation’ section

The details of the ATR-FTIR instrument should better be described in this manuscript.

4. In ‘Pre-processing of spectral data’ section

Normalization was used as the data pre-processing in this research, and as described in Line552-555, ‘If a non-normalized spectral dataset was used, the discrimination done by the machine learning models would primarily be based on the absorbance intensity’. However, is there statistically significant difference of spectral intensities between malignant and benign samples, would it be eliminated by the Normalization procedure?

Had other efforts for eliminating the systematic error besides normalization been ever tried in this filed of researches, such as introducing internal standard compound in the ATR-FTIR analysis or using the quantitative results of cancer related molecules obtained by FTIR as the inputs rather than the relative intensities of absorption peaks?

5. In ‘Results, Diagnostic performance of machine learning models’ section

According to the results the authors provided, and also as they described in line 570, the advantage of NN models compared to SVM model is not obvious, thus the superiority of NN models in ATR-FTIR data-based cancer diagnosis should be further discussed. The question that why NNs were recommended is not well answered.

6. In ‘Discussion’ section

Since this manuscript is focused on the algorithm optimization and comparison, please consider that is it necessary that the relations between IR absorption peaks and pathological process of cancer tissues were discussed as in detail as in the manuscript, especially when the spectra and information about the specimens were not provided.

6. PLOS authors have the option to publish the peer review history of their article (what does this mean?). If published, this will include your full peer review and any attached files.

Reviewer #1: No

Reviewer #2: No

---

## [Author Response · Author response to Decision Letter 0]

24 Oct 2021

Our response to the reviewers are better read in the docx file provided upon submission. 

We would like to thank the editor and the two reviewers for the careful reading of our manuscript and for their constructive questions and insightful comments. Our responses can be found highlighted in italic letters and the revisions made in the manuscript are in blue font. 

Editorial requests:

Comment:

The reviewers offer a number of technical critiques that I believe are best handled through a major revision. In particular, one point that both reviewers agree on is the viability of the normalization methods used. This caught my eye as well, and I recommend that the authors consider more traditional normalization methods (such as peak normalization) or justify their use of the proposed methods described in Equation 1. In addition, Reviewer 2 brings up some more practical experimental questions that may require significant changes in data analysis that will require additional review.

Our answer: 

We thank the editor for the above comment. The manuscript has been edited according to the reviewers’ comments. Moreover, we have included brief responses for the reviewers’ questions.

Reviewer Comments:

Reviewer #1: 

This paper is aimed to develope neural network (NN) models to discriminate malignant lung cancer samples from benign ones. 5 different types of NN models have been developed and the performance of these NNs were compared to the performances of 6 different artificial intelligence (AI) methods. CNNs were found to perform best and the nearest AI method was SVM. I have the following comments to improve the manuscript:

Comment 1: Preprocessing of spectral data: The preprocessing was done by rubber-band baseline correction followed by mean centering (to 0) and std normalization (to 1) using Eq.1. However, these two steps will not be sufficient to make the spectral data to be independent from the equipment, operator, or time of acquisition. (To make this more clear consider an exaggerated case for scaling operation: Assume 100 samples of spectra 90 of which have almost the same appearance but 10 spectra are taken with a gain factor of about 0.1. Now, if the method in the manuscript is used, after preprocessing, the 10 low-amplitude spectra will have high negative values because mean spectra are dominated by high amplitude x-variables. The score values in the PCA plot for these low amplitude spectra will be always highly negative along the principal component F1. This may not have been the case. Multiplying, for this case, the low amplitude samples by 10 may have resulted by in positive values for the first principal component.) I would suggest either unit vector normalization or peak normalization to the baseline corrected spectra before applying the scaling in Eq.1.

Our answer:

We thank the reviewer for the above comment. We awknowledged the use of the unit vector normalization instead of the z-score normalization in performing the study’s peak analysis (Table 3) and in illustrating the sample spectrum (Figure 4); since such normalization would not yield negative values, for the absorbances, which may be baffling to future readers. However, we no longer saw the need to change our normalization in training our models since the two normalization processes (z-score and unit vector) are similarly preferable for neural networks; the informal proof is provided in the "Response to Reviewers" docx file.

Thus, lines 184-206 of pages 8-9 concerning the pre-processing of spectral data now read:

The spectral dataset X_SD consists of 122 elements, of which corresponds to 66 labeled benign samples, and 56 malignant samples. Each element X_SD^((i))∈X_SD in the spectral data set consists of 462 variables, which corresponds to the absorbance of a tissue sample for each wavenumber from 1800 cm-1 to 850 cm-1.

Normalization and baseline correction comprised the pre-processing of the spectral data set, X_SD. All spectral data was normalized using unit vector normalization before peak analysis and spectrum visualization. The normalization was performed to eliminate bias from y-value discrepancies among the IR samples, as well as to maintain an average minimum absorbance of 0 for each spectral data. The unit vector normalization equation is shown as follows

Refer to the "Response to Reviewers" docx file (1)

With regards to the training process, however, the spectral data was normalized using z-score normalization instead, since it is the recommended method of normalization for the scaled exponential linear unit (SELU)-based feed forward neural networks [34]. The normalization was done per spectral vector using the equation

Refer to the "Response to Reviewers" docx file (2)

where the mean(X_SD^((i)) ) and std (X_SD^((i))) notations denote the mean and standard deviation of the elements of the vector X_SD^((i)).The implemented normalization scales the elements of X_SD^((i)) to have an overall mean of 0 and a standard deviation of 1. Note that similar to the unit vector normalization, the z-score normalization was performed per spectral vector X_SD^((i)).

Comment 2: What were the inputs to the SVM classifier? (First two principal components of the PCA scores, or more?) To determine the SVM classifier performance would full-cross validation (Leave -one-out) be more preferrable or reliable?

Our answer: We thank the reviewer for the above comment. Inputs are not the first two principal components but rather the whole spectrum (462 variables). We believe that the 10-fold cross validation will be enough if not better than the full-cross validation leave one out method. On the performed validation the data set was divided into 10 mutually exclusive sets in which a model was trained for each pass for each fold. Moreover, the 10-fold cross validation was performed 50 times. Hence, 500 models are effectively trained using different data sets. Considering this number of trials, we think this would suffice for reliability.

Comment 3: Several papers appeared in literature recently about the use of infrared spectroscopy in lung cancer characterization and differential diagnosis using different sample types obtained from human individuals. For example, see Abbas et al. 2018, J. Biomedical Optics and Kaznowska et al., 2018, Talanta. Brief discussion and comparison are necessary between the results of the present and previously published studies.

Our answer:

We thank the reviewer for the above comment and for recommending articles for further reading. We have read the suggested articles and decided to add them to the discussion of our results, particularly the results of Kaznowska et al., 2018 which can be found in lines 739-741 of page 37:

“The peak absorbances for glycogen (1051 cm-1) and phosphorylated proteins (880 cm-1) were significantly increased in malignant samples. These are aligned with the findings of several lung cancer studies [40,48,79,80], but contrary to the findings of Kaznowska et al. where glycogen is decreased in squamous cell carcinoma and adenocarcinoma [64]” 

and in lines 818-830 of pages 40-41:

“There is a potential for the improvement of lung cancer diagnosis by introducing an automated method of classifying tissues based on the vibrational spectroscopic patterns in the infrared fingerprint region. A similar study done by Kaznowska et al. showed that FTIR spectral data from distinct regions can be used to discriminate between benign and malignant lung cancer tissue using PCA-LDA and a physics-based computation model to analyze their data [64]. Their results showed that the model is 78% to 99% sensitive and 65% to 99% specific [64]. To further advance these findings, more sophisticated computational tools such as NNs may be used in the discrimination of spectral data. The designed NNs are 89.84% to 92.87% sensitive (RR) and 86.63% to 96.01% specific (SR). While the sensitivity and specificity of the earlier study can reach 99%, it has higher variability in discriminating samples compared to the NNs. Thus, SVM or NN-aided diagnosis using infrared spectroscopy may be utilized and is a better adjunct tool in identifying truly benign or malignant tissue samples.”

However, Abbas et al., 2018 utilized pleural fluid samples to discriminate malignant pleural mesothelioma from lung cancer and benign pleural effusion. We believe that it cannot be compared to our results because of the difference in objective and specimen used. 

 

Reviewer #2: 

A research about the application of machine learning combined with ATR-FTIR in the lung cancer diagnosis is described in the manuscript. The experiments and data processing were comprehensively designed and conducted, the description is detailed and the statistics of data is basically appropriate. Although it is Not a highly innovative idea, this research and the article did give a good practice on presenting the potential of NN models and even other ML methods as an effective tool in diagnosing lung cancer based on ATR-FTIR spectra. Thus, this manuscript could be accepted after a major revision. The following issues need to be further considered and explained.

Comment 1: In ‘Introduction’ section, the evaluation of radiography methods compared to the FTIR method in line 26 is not completely reasonable and just. The FTIR method presented in this article is based on invasive sampling operation, whether it is suitable for routine health check rather than the non- invasive LDCT method? Besides, the limited and fully evaluated risk of exposure to radiation and the issue of the time and the workload demand could not be accounted as the main disadvantages of LDCT test. Please reconsider that is this part of statement fully appropriate for publication.

Our answer: 

We thank the reviewer for the above comment. We have edited the manuscript following the recommendations for the introduction. Thus, lines 65-86 of pages 2-3 which explain the diagnostic method of lung cancer now read:

“The initial evaluation of patients suspected of lung cancer starts with history taking and physical examination complemented with complete blood count and chest radiography [3,4]. A negative result from the chest radiography, however, is not definitive as the location and size of the tumor, state of metastasis, and type of lung cancer must be checked as well [3,5]. All patients who are eligible for treatment with the intention of curing the disease must be offered with computed tomography (CT) [4,6], a positron emission tomography (PET) scan if necessary, and then a diagnostic evaluation shall be made [3]. CT scans must be done prior to any invasive procedures as it provides knowledge regarding anatomical changes which increase the diagnostic yield of investigation [4]. At present, Centers for Disease Control (CDC) only recommends low-dose computed tomography (LDCT) as the means of screening lung cancer [7]. However, this process is still known to provide false-positive results, leading to overdiagnosis [7], while also exposing the patients to low doses of radiation [7,8]. 

Combinations of other testing are also employed. Magnetic resonance imaging (MRI), bronchoscopy, and histopathologic examinations are done as needed for diagnosis [9]. Although MRI is known to offer a non-invasive assessment without the radiation, it is still susceptible to cardiac and respiratory motion artifacts [9]. Meanwhile, bronchoscopy plays an important role in confirming diagnosis but it is dependent on tumor size and location [4]. The current gold standard, which is the microscopic examination of hematoxylin and eosin (H&E)-stained biopsies, would take about a week to complete [10]. Additionally, it is prone to interobserver variability, leading to diagnostic disagreement among pathologists which may affect the prognosis and future course of action [11–16]. In light of a disagreeing diagnosis, a genetic and/or molecular analysis may be requested for an even more definite diagnosis and treatment guidance [17].”

Additionally, we have removed the comparison of LDCT and FTIR through biopsy samples. Thus lines 96-100 of page 4 now read: 

“Ergo, an axillary tool that has the combined characteristics of the aforementioned auspicious diagnostic measurement and the latest technological update may be utilized that would give birth to advancement in medical diagnostics. This would not only relieve the workload of medical professionals but could also provide a tool that is even more specific and sensitive with a shorter turn-around time.”

Comment 2: In ‘Materials and methods, Study population and sample preparation’ section Although the specimen processing and FTIR analysis have been reported in previous publication, it would be helpful for understanding this research and could also make it easier to read if these details were described in this manuscript, including issues such as the size of the samples, the sampling operation and the pretreatment of the specimens before the FTIR analysis, and whether the FTIR spectra were obtained from a small sampling point or as an average result of the entire specimen. Besides, the optical images of the specimens, graphical illustration of the FTIR analyzing process and the FTIR spectra were preferred in the manuscript or at least in Supplementary materials.

Our answer:

We thank the reviewer for the valuable comment to explain in detail how the specimen processing and FTIR analysis were done in the previous study. Hence, we have edited the manuscript following the reviewer’s recommendations. The revised ‘Materials and methods, Study population and sample preparation’ section, in lines 140-152 of page 6 now reads:

“FTIR spectral data (n =122; 56 malignant, 66 benign) of FFPE lung biopsies from adult patients seen at MMMH-MC and USTH from 2015 to 2017 comprised the dataset. No participants were recruited for this study as the spectral data were acquired from the previous study of Bangaoil et al. [21]. 

Sample preparation and pretreatment of specimens before ATR-FTIR analysis were also discussed in the previous study [21]. Three (3) adjacent sections were cut uniformly (5-μm thick) from the FFPE cell blocks using a microtome (Leica Biosystems, Germany) and then mounted on glass slides. The outer sections (2) were stained with H&E and distributed to two (2) external evaluators (pathologists) blinded of the original diagnosis for validation. The middle section was deparaffinized following standard protocols using xylol [31,32], washed with water, and left to air dry overnight before spectral analysis [33]. The classification (i.e., whether spectral data was benign or malignant) was based on the microscopic examination of H&E-stained tissues from each study site.”

Furthermore, we would like to clarify that the ATR-FTIR spectrometer used in the study has no capacity to provide infrared optical images of the specimens (model is further discussed in comment 3). To better illustrate how the spectral data were obtained from both benign and malignant samples, lines 153-164 of page 7 were added to the revised manuscript:

“Prior to spectral measurement, performance qualification (PQ) test protocols using the automated validation program of the OPUS 8.0 software were conducted. The deparaffinized tissue sections were placed and oriented directly in contact with the ATR diamond surface (2 mm x 2 mm). All tissue sections were examined in the mid-IR region (4000 cm-1 to 600 cm-1) with an average of 48 scans, yielding spectral data with a resolution of 4 cm-1. For the H&E stained malignant sections, the majority were entirely scanned since the sections were completely composed of cancer cells. A small number of malignant sections that were identified by pathologists to only have areas with concentrated cancer cells were scanned in the indicated area. On the other hand, benign tissue sections were scanned randomly, covering 50% of the total area of the section. All tissue sections were scanned three (3) times for reproducibility. The spectral data of all sections in the fingerprint region (1800 cm-1 to 850 cm-1) were extracted and the median infrared spectra of both malignant and benign samples in the said region were calculated.”

Moreover, as recommended, a graphical illustration (figure 1) which shows a summarized version of the method that was implemented in the study was added. Thus, lines 163-167 of page 7 now read:

“The overall method implemented in the study is summarized in Figure 1, through a generalized process flowchart.”

Fig 1 was added as follows:

Fig 1. Experimental design process flowchart. The figure shows the experimental design implemented in the study, from the acquisition of FFPE lung tissue samples, to the machine learning training and evaluation.

Lastly, we have also included the sample spectrum in the results section (figure 5) to further present the difference in absorbance spectra of malignant and benign classes. 

Fig 5 was added in lines 438-443 of page 22:

Fig 5. Median ATR-FTIR absorbance spectra of malignant (n = 53) and benign (n = 65) lung tissue samples. The figure shows the median FTIR spectrum of malignant and benign lung tissue samples and their corresponding peaks identified via visual analysis.

Comment 3: In ‘Data measure/instrumentation’ section, The details of the ATR-FTIR instrument should better be described in this manuscript.

Our answer:

We thank the reviewer for this suggestion to add the details of the ATR-FTIR instrument that was used. Thus, lines 170-174 of page 7 of the revised manuscript now read:

“Bruker Alpha II Fourier Transform Infrared (FTIR) spectrometer (Bruker Optics, Germany) equipped with a platinum ATR single reflection monolithic diamond sampling module was utilized to acquire the infrared (IR) spectra. The fully automated validation program of OPUS 8.0 software (Bruker Optics, Germany) was used for the performance qualification (PQ) test. The said software was also used for baseline correction of obtained IR spectra.”

Comment 4: In ‘Pre-processing of spectral data’ section Normalization was used as the data pre-processing in this research, and as described in Line 552-555, ‘If a nonnormalized spectral dataset was used, the discrimination done by the machine learning models would primarily be based on the absorbance intensity’. However, is there statistically significant difference of spectral intensities between malignant and benign samples, would it be eliminated by the Normalization procedure? Had other efforts for eliminating the systematic error besides normalization been ever tried in this field of researches, such as introducing internal standard compound in the ATR-FTIR analysis or using the quantitative results of cancer related molecules obtained by FTIR as the inputs rather than the relative intensities of absorption peaks?

Our answer:

Again, we thank the reviewer for the above comment. The statistical significance from the peak analysis is affected by the normalization. However, the problem of not normalizing is that FTIR would generalize to recognize benign and malignant cells by the sample’s density rather than their composition.

We have edited the manuscript following the Editor’s recommendations. The difference between using a normalized and a non-normalized data set was mentioned in page 38 lines 763-766, which reads:

“If a non-normalized spectral dataset was used, the discrimination done by the machine learning models would primarily be based on the absorbance intensity, whereas the use of normalized spectra would highlight the distinct biochemical structures that could differentiate malignant from benign samples [85]”

Furthermore,…..

A performance qualification test was conducted by the team of Bangaoil et. al prior to spectral measurement to reduce and eliminate possible systematic errors. This included a series of tests that the samples must pass before ATR-FTIR analysis can be performed. These tests include the signal-to-noise test, deviation from 100%-line test, interferogram peak test, and wavenumber accuracy test. Additionally, before the samples were scanned, the background spectrum was recorded and systematically subtracted by the software thus eliminating atmospheric effects. 

Thus, lines 208-210 of page 9 of the revised manuscript now reads:

“Prior to the acquisition of spectral data, baseline correction and performance qualification tests had been conducted in the previous study [21]. The performance qualification tests included the signal-to-noise test, deviation from 100%-line test, interferogram peak test, and wavenumber accuracy test, all of which must be passed before ATR-FTIR analysis can be performed.”

Comment 5: In ‘Results, Diagnostic performance of machine learning models’ section According to the results the authors provided, and also as they described in line 570, the advantage of NN models compared to SVM model is not obvious, thus the superiority of NN models in ATR-FTIR data-based cancer diagnosis should be further discussed. The question that why NNs were recommended is not well answered.

Our answer:

Again, we thank the reviewer for the above comment. The notion of NN models being superior compared to the SVM model was further expounded in the discussion section based on the diagnostic metrics – including the corresponding clinical implication of having a high value for a particular metric. 

Thus, lines 776-799 of pages 38-39 of the revised manuscript now read:

“Among the classical models, only SVM had performance metrics comparable to that of NNs. The significantly higher ACC rate of the NNs compared to SVM denotes that NNs are generally better overall classifiers. ACC is a viable intuitive metric of comparison since the spectral dataset used in the study was roughly balanced (53 malignant and 65 benign), and both classes (i.e., malignant and benign) are considered of equal importance. However, such metric does not entail better individual class effectiveness [88]. The non-significantly different SR values of most NNs and SVM indicate the equal effectiveness of the models in identifying benign spectra. Meanwhile, the RR values of NNs were significantly lower than that of SVM, implying that the latter is more effective in identifying malignant spectra.

In terms of predictive values, there were no significant differences in the NPVs of NNs and SVM, signifying equal predictive power in identifying truly benign samples. Conversely, the majority of NNs were observed to have significantly higher PPVs compared to SVM. Such denotes that most NNs were better classifiers in terms of identification of malignant spectra as truly malignant. A high PPV reduces the occurrence of false positives [89]. Falsely classifying patients as having malignant tumors can result in negative short-term psychosocial consequences [90], possibly resulting in non-adherence to subsequent lung cancer screening tests [91]. Moreover, misdiagnosing patients with rare benign tumors that mimic malignant neoplasm will not only entail higher medical care costs [92], but will also lead to an intractable condition due to further surgical procedures, chemotherapy, and radiotherapy that are designed for treating malignant cases [93]. 

The choice between NNs and SVM is highly dependent on the point of interest presented by the problem domain. If there is a need for more accurate true malignant infrared spectra detection, NNs, specifically CNN, may be deemed more superior than SVM. Conversely, for a more accurate true benign infrared spectra detection, either NNs or SVM may be utilized since both were observed to be comparable.”

Comment 6: In ‘Discussion’ section, Since this manuscript is focused on the algorithm optimization and comparison, please consider that is it necessary that the relations between IR absorption peaks and pathological process of cancer tissues were discussed as in detail as in the manuscript, especially when the spectra and information about the specimens were not provided.

Our answer:

Again, we thank the reviewer for the above comment. Although the manuscript is focused on the algorithm optimization and comparison, the relation between absorption peaks and pathological process of cancer tissues is deemed necessary. Since the goal of the manuscript is to explore the potential of ATR-FTIR and AI in the diagnosis of lung cancer, providing a clinical perspective regarding the significant differences in the spectral data of each class is essential in possibly explaining how the NNs came up with the classification for each sample.

---

## [Decision Letter · Decision Letter 1]

31 Jan 2022

PONE-D-21-23770R1Artificial neural network in the discrimination of lung cancer based on infrared spectroscopyPLOS ONE

Dear Dr. Lugtu,

Thank you for submitting your manuscript to PLOS ONE. After careful consideration, we feel that it has merit but does not fully meet PLOS ONE’s publication criteria as it currently stands. Therefore, we invite you to submit a revised version of the manuscript that addresses the points raised during the review process.

 The main points of contention from the previous reviews are the missing citation from Abbas. If the authors still disagree with the citation, I highly recommend justification with respect to the new comments by Reviewer 1. In addition, Reviewer 3 brings up some new points regarding data processing and consistency that are important to address.

We look forward to receiving your revised manuscript.

Kind regards,

David Mayerich

Academic Editor

PLOS ONE

Journal Requirements:

Reviewers' comments:

Reviewer's Responses to Questions

**Comments to the Author**

1. If the authors have adequately addressed your comments raised in a previous round of review and you feel that this manuscript is now acceptable for publication, you may indicate that here to bypass the “Comments to the Author” section, enter your conflict of interest statement in the “Confidential to Editor” section, and submit your "Accept" recommendation.

Reviewer #1: (No Response)

Reviewer #3: (No Response)

2. Is the manuscript technically sound, and do the data support the conclusions?

Reviewer #1: Yes

Reviewer #3: Yes

3. Has the statistical analysis been performed appropriately and rigorously? 

Reviewer #1: Yes

Reviewer #3: Yes

4. Have the authors made all data underlying the findings in their manuscript fully available?

Reviewer #1: Yes

Reviewer #3: No

5. Is the manuscript presented in an intelligible fashion and written in standard English?

Reviewer #1: Yes

Reviewer #3: Yes

6. Review Comments to the Author

Reviewer #1: I am satisfied with the answers to comments 1 and 2. However, as regard to the answer for comment 3 I have some concern.

My main concern is that, some recent previous works on lung cancer diagnosis are missing in this manuscript. The authors responded to comment 3 that “Abbas et. al. 2018 should not be included into the references because they utilized pleural fluid samples to discriminate malignant pleural mesothelioma from lung cancer and benign pleural effusion. We beleive that it cannot be compared to our results because of the difference in objective and specimen used”.

However, in the introduction part of the manuscript there are references related to thyroid cancer,

ovarian cancer, breast cancer, image base diagnosis,and some infrared papers on lung cancer etc.

which have different objectives and specimens.

On the other hand, when you examine Abbas et al’s paper not only mesothelioma samples are

discriminated from lung cancer, but in addition, lung cancer samples are discriminated from both

benign samples and mesothelioma samples using unsupervised and supervised methods.

(please see Fig. 3 of that paper). Therefore the recent paper of Abbas et al. J. Biomed. Optics,

2018 paper should be cited

Reviewer #3: Finding optimal NN model and optimal parameters of NN by using grid search with GA on spectroscopic data is the main idea of this manuscript which has been appropriately implemented here. As per the manuscript, data consists of 122 spectral vectors; it is not mentioned how these spectra are collected. Are these collected from 122 different patients/tissue blocks? If each spectral belongs to a separate tissue block, how did you select signal spectral vector to represent the entire benign tissue or tumor part of malignant tissue? Did you collect more spectra and then take the average? With FTIR spectral collection, spectral variance is observed depending on many factors like underlying histology class in the tissue or sample density at that location.

I agree with the other reviewers that instrument parameters or environmental factors can affect spectral profile; hence proper pre-processing like baseline correction and peak normalization (not unit normalization) is needed.

Also, for some classifiers, data with entire variables are used, and PCA components are used for others. Although it should not have a big difference in the results, it is better to have similar input to all the classifiers while comparing the performances. Here, the number of variables/features is way more than the number of training samples; hence generalization of the classifier performance is not guaranteed.

You can either remove equations for classic classifiers or write them correctly. Text at line 269 is not very clear; it does not properly connect with the inline equation 7. Also, in line 277, the authors talk about equations but instead provide references. Again, the authors discuss terms from these equations which are not present in the manuscript. Again at line 280, α is explained without any mention of α before that.

The manuscript makes it hard to understand what you mean by individuals in GA; please describe it explicitly.

7. PLOS authors have the option to publish the peer review history of their article (what does this mean?). If published, this will include your full peer review and any attached files.

Reviewer #1: No

Reviewer #3: No

---

## [Author Response · Author response to Decision Letter 1]

16 Mar 2022

All responses to the reviewers are comprehensively stated in the docx file included in this submission. 

Again, thank you for giving us a chance to revise our manuscript for PLOS ONE.

---

## [Decision Letter · Decision Letter 2]

28 Apr 2022

Artificial neural network in the discrimination of lung cancer based on infrared spectroscopy

PONE-D-21-23770R2

Dear Dr. Lugtu,

We’re pleased to inform you that your manuscript has been judged scientifically suitable for publication and will be formally accepted for publication once it meets all outstanding technical requirements.

Kind regards,

David Mayerich

Academic Editor

PLOS ONE

Additional Editor Comments (optional):

Reviewers' comments:

Reviewer's Responses to Questions

**Comments to the Author**

1. If the authors have adequately addressed your comments raised in a previous round of review and you feel that this manuscript is now acceptable for publication, you may indicate that here to bypass the “Comments to the Author” section, enter your conflict of interest statement in the “Confidential to Editor” section, and submit your "Accept" recommendation.

Reviewer #1: All comments have been addressed

Reviewer #3: (No Response)

2. Is the manuscript technically sound, and do the data support the conclusions?

Reviewer #1: Yes

Reviewer #3: Yes

3. Has the statistical analysis been performed appropriately and rigorously? 

Reviewer #1: Yes

Reviewer #3: Yes

4. Have the authors made all data underlying the findings in their manuscript fully available?

Reviewer #1: Yes

Reviewer #3: No

5. Is the manuscript presented in an intelligible fashion and written in standard English?

Reviewer #1: Yes

Reviewer #3: Yes

6. Review Comments to the Author

Reviewer #1: The revised manuscript is technically sound and all the comments raised by the reviewer were taken into consideration

Reviewer #3: Revised manuscript looks good. However, figures are not self-explanatory, input/output format to DNN models are not shown in the figures. Also, adding model summary can help other to reuse the proposed models. Figure 9 is hard to understand, even description in the main text is not very clear. Figure captions should be standalone, i.e., descriptive enough to be understood without having to refer to the main text.

7. PLOS authors have the option to publish the peer review history of their article (what does this mean?). If published, this will include your full peer review and any attached files.

Reviewer #1: No

Reviewer #3: **Yes: **Rupali Mankar

---

## [Editor Report · Acceptance letter]

2 May 2022

PONE-D-21-23770R2 

Artificial neural network in the discrimination of lung cancer based on infrared spectroscopy 

Dear Dr. Lugtu:

I'm pleased to inform you that your manuscript has been deemed suitable for publication in PLOS ONE. Congratulations! Your manuscript is now with our production department. 

Kind regards, 

on behalf of

Dr. David Mayerich 

Academic Editor

PLOS ONE